# Don't Trade Off Safety: Diffusion Regularization for Constrained Offline RL

**Junyu Guo**[*]
University of California, Berkeley

**Zhi Zheng**
University of California, Berkeley

**Donghao Ying**
University of California, Berkeley

**Ming Jin**
Virginia Tech

**Shangding Gu**
University of California, Berkeley

**Costas Spanos**
University of California, Berkeley

**Javad Lavaei**
University of California, Berkeley

## Abstract

Constrained reinforcement learning (RL) seeks high-performance policies under safety constraints. We focus on an offline setting where the agent learns from a fixed dataset—a common requirement in realistic tasks to prevent unsafe exploration. To address this, we propose Diffusion-Regularized Constrained Offline Reinforcement Learning (DRCORL), which first uses a diffusion model to capture the behavioral policy from offline data and then extracts a simplified policy to enable efficient inference. We further apply gradient manipulation for safety adaptation, balancing the reward objective and constraint satisfaction. This approach leverages high-quality offline data while incorporating safety requirements. Empirical results show that DRCORL achieves reliable safety performance, fast inference, and strong reward outcomes across robot learning tasks. Compared to existing safe offline RL methods, it consistently meets cost limits and performs well with the same hyperparameters, indicating practical applicability in real-world scenarios. We open-source our implementation at `https://github.com/JamesJunyuGuo/DRCORL`.

## 1 Introduction

Offline reinforcement learning (RL) has advanced decision-making by learning from pre-collected datasets without online interaction [Fujimoto et al., 2019, Levine et al., 2020]. For real-world control tasks (e.g., autonomous driving, industrial control), safety is equally critical. Safe RL addresses this by imposing constraints, often formulated as a constrained Markov decision process (CMDP) [Gu et al., 2024c, Altman, 2021], to ensure high performance without violating safety requirements. These can be hard constraints (no violation at each step) [Zheng et al., 2024, Ganai et al., 2024] or soft constraints (expected total cost below a threshold) [Chow et al., 2018, Yang et al., 2020, Koirala et al., 2024, Guo et al., 2025, Shi et al., 2021]. We focus on the soft-constraint setting in this work.

Offline safe RL faces two main hurdles: distribution shift and reward-safety trade-off. Firstly, the learned policy may deviate from the offline dataset's state–action distribution, causing critic overestimation and extrapolation errors [Fujimoto et al., 2019, Lyu et al., 2022]. To address value overestimation, previous methods have either constrained the learned policy to remain close to the behavioral policy Wu et al. [2019], Kumar et al. [2019] or conservatively penalized over-optimistic out-of-distribution (OOD) state-action pairs Kostrikov et al. [2021], Lyu et al. [2022], Xu et al. [2022].

---

[*]Corresponding authors: `junyuguo24@berkeley.edu`, `shangding.gu@berkeley.edu`

Secondly, achieving high returns while strictly respecting safety becomes more challenging when these objectives conflict. Although constrained optimization methods [Achiam et al., 2017, Liu et al., 2022] handle this in online RL, they rely on on-policy data collection, making them not directly applicable to offline settings. Hence, the key question is:

> *How can we balance reward maximization and constraint satisfaction without risking out-of-distribution actions or unsafe behavior in a setting where no additional data can be collected?*

To address this, we propose *Diffusion-Regularized Constrained Offline Safe Reinforcement Learning* (DRCORL). First, DRCORL trains a diffusion policy to imitate the behavioral policy in the offline dataset; then it regularizes the learned policy via the diffusion model's score function—removing the need for costly sampling from the diffusion model at inference. Second, we apply gradient manipulation to balance reward optimization and cost minimization, effectively handling conflicts between these two objectives. Furthermore, the behavioral policy serves as a regularizer, discouraging OOD actions that may compromise safety. While prior works have independently explored generative policy regularization and constraint-aware optimization, they typically treat these aspects as orthogonal. Our framework unifies them by showing that diffusion-based behavior modeling naturally induces a trust-region–like regularization that complements gradient-level safety adaptation. This conceptual bridge highlights diffusion modeling not merely as a behavioral prior, but as a principled mechanism for safety-constrained learning.

We evaluate DRCORL on the DSRL benchmark [Liu et al., 2023a], comparing against state-of-the-art offline safe RL methods. Experiments show that DRCORL consistently attains higher rewards while satisfying safety constraints. Our main contributions are: ① We exploit diffusion-based regularization to build a simple, high-speed policy with robust performance; and ② We introduce a gradient-manipulation mechanism for reward–cost trade-offs in purely offline settings, ensuring safety without sacrificing returns.

## 2 Preliminary

A Constrained Markov Decision Process (CMDP) [Altman, 2021] is defined by the tuple $\langle \mathcal{S}, \mathcal{A}, P, r, c, \gamma \rangle$, where $\mathcal{S}$ and $\mathcal{A}$ represent the state and action spaces, respectively. The transition kernel $P$ specifies the probability $P(s'|s,a)$ of transitioning from state $s$ to state $s'$ when action $a$ is taken. The reward function is $r : \mathcal{S} \times \mathcal{A} \to \mathbb{R}$, and the cost function is $c : \mathcal{S} \times \mathcal{A} \to \mathbb{R}$. The discount factor is denoted by $\gamma$. A policy is a function $\pi : \mathcal{S} \to \Delta(\mathcal{A})$ that represents the agent's decision rule, i.e., the agent takes action $a$ with probability $\pi(a|s)$ in state $s$, and we define $\Pi$ as the set of all feasible policies. Under policy $\pi$, the value function is defined as $V_\diamond^\pi(\rho) = \mathbb{E}[\sum_{t=0}^\infty \gamma^t \diamond (s_t, a_t)|s_0 \sim \rho]$, where $\diamond \in \{r, c\}$, $\rho$ is the initial distribution, and the expectation is taken over all possible trajectories. Similarly, the associated Q-function is defined as $Q_\diamond^\pi(s,a) = \diamond(s,a) + \gamma \mathbb{E}_{s' \sim P(\cdot|s,a)}[V_\diamond^\pi(s')]$ for $\diamond \in \{r, c\}$. In standard CMDP, the objective is to find a policy $\pi \in \Pi$ that maximizes the cumulative rewards $V_r^\pi(\rho)$ while ensuring that the cumulative cost $V_c^\pi(\rho)$ remains within a predefined budget $l$.

In the offline setting, the agent cannot interact directly with the environment and instead relies solely on a static dataset $\mathcal{D}^\mu = \{(s_i, a_i, r_i, s'_i, c_i)\}_{i=1}^N$ consisting of multiple transition tuples, which is collected using a behavioral policy $\pi_b(a|s)$. This offline nature introduces the risk of distributional shift between the dataset and the learned policy. To mitigate this, an additional constraint is often imposed to limit the deviation of the learned policy $\pi$ from the behavioral policy $\pi_b$, resulting in the optimization problem

$$\max_{\pi \in \Pi} \mathbb{E}[V_r^\pi(\rho)] \text{ s.t. } D_{\mathrm{KL}}(\pi \| \pi_b) \leq \epsilon, \mathbb{E}[V_c^\pi(\rho)] \leq l, \tag{1}$$

where $D_{\mathrm{KL}}(p\|q)$ the KL-divergence between two distributions defined as $D_{\mathrm{KL}}(p\|q) = \mathbb{E}_{x \sim p}[\log(p(x)/q(x))]$. We use the KL-divergence to penalize the learned policy $\pi$'s distance to the behavioral policy, though it is actually not a distance measure. To address safety constraints, primal-dual methods Ding et al. [2021], Paternain et al. [2022], Wu et al. [2024] typically reformulate the constrained optimization problem as follows:

$$\max_{\pi \in \Pi} \mathbb{E}[V_r^\pi(\rho)] - \lambda(\mathbb{E}[V_c^\pi(\rho)] - l) \text{ s.t. } D_{\mathrm{KL}}(\pi \| \pi_b) \leq \epsilon, \tag{2}$$

where $\lambda \geq 0$ is a surrogate for the Lagrange multiplier. When the safety constraint is violated, the multiplier $\lambda$ increases to impose a greater penalty on the cost, thereby reducing the cost value.

## 3 Methodology

### 3.1 Diffusion Model for Policy Extraction

Our main idea is to fully exploit the offline dataset to obtain a behavioral policy, and use the behavioral policy to guide the training of the target policy. A policy $\pi(a|s)$ is a distribution on the action space. Previous work estimated the behavior policy with maximum likelihood estimation Fujimoto and Gu [2021] or leveraged a conditional variational autoencoder Kingma [2013], Sohn et al. [2015]. Here, we exploit the concept of diffusion models Sohl-Dickstein et al. [2015], Ho et al. [2020], Song et al. [2020b] to learn the unknown behavioral policy $\pi_b(a|s)$ given its strong generative capabilities. Diffusion models have emerged as powerful generative tools to generate data sample $x_0 \sim p(x_0)$ with few-shot samples. They work by using a forward process $q(x_t|x_0)$ to perturb the original distribution to a known noise distribution. Subsequently, this model generates the samples using the reverse denoising process $p_\psi(x_{t-1}|x_t)$. The forward process can generally be written with a forward stochastic differential equation (SDE)

$$dx = -\frac{\beta(t)}{2}xdt + \sqrt{\beta(t)}dW_t, \tag{3}$$

where $\beta(\cdot) : [0, T] \rightarrow \mathbb{R}^+$ is a scalar function and the process $\{W_t\}_{t \in [0,T]}$ is a standard Brownian motion. Our forward process is the discretized version of SDE in Eq. (3) perturbing the original distribution to Gaussian noise. For example, if we choose a variance preserving SDE for the forward diffusion process as in Ho et al. [2020], each step $x_t$ is perturbed with the noise $z_t \sim \mathcal{N}(0, I)$ to obtain $x_{t+1} = \sqrt{\alpha_t}x_t + \sqrt{\beta_t}z_t$, where $\beta_t = 1 - \alpha_t \in (0, 1)$. We denote $\bar{\alpha}_t = \prod_{i=1}^t \alpha_i$ and $\bar{\beta}_t = 1 - \bar{\alpha}_t$. Therefore, we can rewrite $x_t = \sqrt{\bar{\alpha}_t}x_0 + \sqrt{\bar{\beta}_t}\epsilon_t$, where $\epsilon_t \sim \mathcal{N}(0, I)$ follows the standard Gaussian distribution. The reverse denoising process is optimized by maximizing the evidence lower bound of the log-likelihood $\mathbb{E}[\log p_\psi(x_0)]$ defined as $\mathbb{E}_{q(x_0:x_T)}\big[\log\big(p_\psi(x_{0:T})/q(x_{1:T}|x_0)\big)\big]$. We can then rewrite the loss function into the weighted regression objective in Eq. (4) and transform the problem into training a denoising function $\epsilon_\psi$ predicting the Gaussian noise $\epsilon_t$:

$$\mathcal{L}(\psi) = \mathbb{E}_{t \in \text{Unif}[0,T]}\big[w(t)\|\epsilon_\psi(x_t, t) - \epsilon_t\|^2\big], \epsilon_t \sim \mathcal{N}(0, I), \tag{4}$$

where $w(t)$ is the weight function. Then the reversing denoising process can be formulated as $x_{t-1} = \frac{1}{\sqrt{\alpha_t}}\big(x_t - \frac{\sqrt{1-\alpha_t}}{\sqrt{1-\bar{\alpha}_t}}\epsilon_\psi(x_t, t)\big) + \sqrt{\beta_t}z_t$, where $z_t \sim \mathcal{N}(0, I)$. Using this notion, we can similarly use the diffusion model to diffuse in the action space $\mathcal{A}$ and sample actions given the current state with the reversing process. In Wang et al. [2022], the authors learned diffusion policies from the offline dataset using guided diffusion. The diffusion policy here is $\pi_\psi(a|s) = \mathcal{N}(a_T; 0, I)\prod_{t=1}^T p_\psi(a_{t-1}|a_t, s)$, where $p_\psi(a_{t-1}|a_t, s)$ is a Gaussian distribution with mean $m_\psi(a_t, t|s)$ and variance $\Sigma_\psi(a_t, t|s)$. See also Lu et al. [2023], Hansen-Estruch et al. [2023], where diffusion policy is used for inference in policy evaluation. The limitation of these methods is that the diffusion models are slow in inference speed, even under the improved sampling scheme in Song et al. [2020a], Lu et al. [2022] the reverse denoising process normally takes at least 10 steps. Therefore, in this work we mainly use diffusion models for pretraining and learning the behavioral policy $\pi_b$. For state-action pair $(s, a)$ in the offline dataset $\mathcal{D}^\mu$, we train our diffusion policy model by minimizing the loss

$$\mathcal{L} = \mathbb{E}_{(s,a) \in \mathcal{D}^\mu}\mathbb{E}_{t \in \text{Unif}(0,T)}\big[w(t)\|\epsilon_\psi(a_t, t|s) - \epsilon_t\|^2\big], \tag{5}$$

where $a_t = \sqrt{\bar{\alpha}_t}a + \sqrt{\bar{\beta}_t}z$ and $z \sim \mathcal{N}(0, I)$. We assume that the diffusion model can perfectly learn the behavioral policy $\pi_b$, as shown in De Bortoli [2022], due to the fact that the target distribution lies on a compact manifold, the first-order Wasserstein distance between the learned policy and the target policy converges to 0 as the discretization length approaches 0.

### 3.2 Diffusion Regularization

The work Chen et al. [2023] showed under the offline RL setting that one can train a simple policy using the pretrained critic and diffusion policy imitating the behavioral policy. The key step is to use the reverse KL divergence to regularize the target policy to be close to the behavioral policy. The forward KL is the KL-divergence $D_{\text{KL}}(\pi^*(\cdot|s)\|\pi_\theta(\cdot|s))$ while the reverse KL is $D_{\text{KL}}(\pi_\theta(\cdot|s)\|\pi^*(\cdot|s))$. As shown in Chen et al. [2023], the forward KL leads to mode covering issue while backward KL encourages mode-seeking behavior, although the latter is harder to optimize. Therefore, in this

work we also choose the reverse KL for regularization. We also constrain our policy family to a simple Gaussian policy class $\Pi : \{\pi_\theta(a|s) = \mathcal{N}(a; m_\theta(s), \Sigma_\theta(s))\}$. Now, we denote the learned diffusion model's score function as $\epsilon_\psi(a_t, t|s)$ and the corresponding diffusion policy as $\mu_\psi(a|s)$. Then, the reverse KL between the policy $\pi_\theta(a|s)$ and the approximated behavioral policy $\mu_\psi(a|s)$ can be written as $-\mathcal{H}(\pi_\theta(\cdot|s)) + H(\pi_\theta(\cdot|s), \mu_\psi(\cdot|s))$. For Gaussian policy, the first part self-entropy term $\mathcal{H}$ can be directly computed in closed form. For $\mathcal{A} = \mathbb{R}^d$, we have

$$\mathcal{H}(\pi_\theta(\cdot|s)) = \int_\mathcal{A} -\log \pi_\theta(a|s)\pi_\theta(a|s)da = \frac{1}{2}\log(\det(\Sigma_\theta(s))) + \frac{d}{2}\log(2\pi) + \frac{d}{2}. \quad (6)$$

Using the reparameterization trick for Gaussian random variables, we can also rewrite the cross entropy term $H(\pi_\theta(\cdot|s), \mu_\psi(\cdot|s))$ as

$$H(\pi_\theta(\cdot|s), \mu_\psi(\cdot|s)) = \mathbb{E}_{\pi_\theta(\cdot|s)}[-\log \mu_\psi(a|s)] = \mathbb{E}_{z \sim \mathcal{N}(0,I)}[-\log \mu_\psi(m_\theta + \Sigma_\theta^{1/2}z|s)]. \quad (7)$$

Finally, to obtain the gradient with respect to the reverse KL divergence we have

$$\nabla_\theta D_{\mathrm{KL}}(\pi_\theta(\cdot|s)||\mu_\psi(\cdot|s)) = \nabla_\theta H(\pi_\theta(\cdot|s), \mu_\psi(\cdot|s)) - \nabla_\theta \mathcal{H}(\pi_\theta(\cdot|s)) \quad (8)$$

$$= \mathbb{E}_{z \sim \mathcal{N}(0,I)}\Big[-\nabla_a \log \mu_\psi(m_\theta(s) + \Sigma_\theta^{1/2}z|s) \cdot \nabla_\theta(m_\theta(s) + \Sigma_\theta^{1/2}z)\Big] - \frac{1}{2}\nabla_\theta \log(\det \Sigma_\theta(s)).$$

In our implementation, the diffusion model serves as a behavioral density estimator, and we use its learned score function $\nabla \log p_\theta(a|s)$ as a tractable surrogate for computing the gradient of the reverse-KL regularization. The work Song et al. [2020b] shows that diffusion models essentially estimate the **score function**:

$$\nabla_x \log p(x_t) \approx s_\psi(x_t, t) = -\frac{1}{\sqrt{\bar{\beta}_t}}\epsilon_\psi(x_t, t). \quad (9)$$

Hence, we can substitute the denoising function $\epsilon_\psi$ into Eq. (8) to directly compute the gradient.

## 3.3 Safe Adaptation

To design our algorithm, we define the reward and cost optimization objectives respectively as follows:

$$\text{Reward:} \max_{\pi \in \Pi} \mathbb{E}_{s \sim \mathcal{D}^\mu}\Big[V_r^\pi(s) - \frac{1}{\beta}D_{\mathrm{KL}}(\pi(\cdot|s)||\pi_b(\cdot|s))\Big] \quad (10)$$

$$\text{Cost:} \max_{\pi \in \Pi} \mathbb{E}_{s \sim \mathcal{D}^\mu}\Big[-(V_c^\pi(s) - l) - \frac{1}{\beta}D_{\mathrm{KL}}(\pi(\cdot|s)||\pi_b(\cdot|s))\Big]. \quad (11)$$

We consider the reverse KL divergence term in Eq. (10) and Eq. (11) as a regularization term to penalize the policy for deviating too far from the behavioral policy $\pi_b$. In practical implementation, we use the learned diffusion policy $\mu_\psi$ to replace $\pi_b$ and compute the gradient for optimization using Eq. (8). As discussed in Gu et al. [2024a], to address the conflict between optimizing reward and cost, it is essential to directly handle the gradient at each optimization step. We also adopt the gradient manipulation idea in Gu et al. [2024a], but we do so under the offline setting where we no longer have access to updating our critic with the on-policy data. The gradient manipulation method can be generally described as follows. For each step where we need to optimize the reward, we update our gradient with $\theta \leftarrow \theta + \eta g_r$, and when we need to optimize the cost our gradient is updated with $\theta \leftarrow \theta + \eta g_c$. The gradient manipulation aims at updating the parameter $\theta$ with a linear combination of the two gradients as $\beta_r g_r + \beta_c g_c$. We can

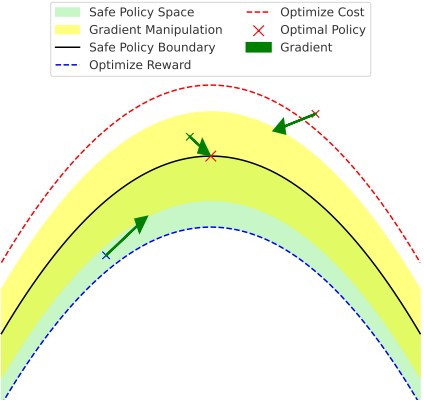

Figure 1: Illustration of the soft switching between safe and unsafe policy regions.

use the angle $\phi := \cos^{-1}\left(\frac{\langle g_r, g_c \rangle}{\|g_r\|\|g_c\|}\right)$ between the two gradients to judge whether the gradients are

conflicting or not, namely if $\phi > 90°$ the two gradients are conflicting and otherwise they are not conflicting. Especially, the final gradient $g$ is computed via the rule

$$g = \begin{cases} \frac{g_r + g_c}{2}, & \text{if } \phi \in (0, 90°) \\ \frac{g_r^+ + g_c^+}{2}, & \text{if } \phi \in [90°, 180°], \end{cases} \tag{12}$$

where $g_r^+ = g_r - \frac{g_r \cdot g_c}{\|g_c\|^2} g_c$ is $g_c$'s projection on the null space of $g_r$ and $g_c^+ = g_c - \frac{g_c \cdot g_r}{\|g_r\|^2} g_r$ is $g_r$'s projection on the null space of $g_c$. In Gu et al. [2024a] under the assumption of the convexity of the optimization target, one can ensure monotonic improvement using gradient manipulation. Therefore we also employ the gradient manipulation method to update our policy. With the procedures outlined above, we present our Algorithm 1, where the safe adaptation step is visualized in Figure 1 and detailed in Algorithm 2 given in Appendix B. Furthermore, we provide the following theoretical results derived from our algorithm, where the proof is deferred to Appendix A.1.

**Proposition 3.1** (Cost Upper Bound). *Assume that the cost function $c : \mathcal{S} \times \mathcal{A} \to [0, c_{\max}]$ is bounded and non-negative. Let $\tilde{\pi}(a|s)$ be the output policy of Algorithm 1. Suppose that there exists $\epsilon_{dist} > 0$ such that $D_{\mathrm{KL}}(\tilde{\pi}(\cdot|s), \pi_b(\cdot|s)) \leq \epsilon_{dist}$ and $D_{\mathrm{KL}}(\pi_b(\cdot|s), \tilde{\pi}(\cdot|s)) \leq \epsilon_{dist}$. Let $\epsilon_{adv}^b = \max_s \mathbb{E}_{a \sim \pi_b(\cdot|s)}[A_c^{\tilde{\pi}}(s, a)]$, where $A_c^\pi(s, a)$ is the advantage function under policy $\pi$, defined as: $A_c^\pi(s, a) = Q_c^\pi(s, a) - V_c^\pi(s)$. Then, it holds that:*

$$V_c^{\tilde{\pi}}(\rho) \leq V_c^{\pi_b}(\rho) + \frac{(c_{\max} + \gamma \epsilon_{adv}^b) \sqrt{2\epsilon_{dist}}}{(1 - \gamma)^2}. \tag{13}$$

This proposition establishes that if the learned policy is constrained to remain within a neighborhood of the dataset's behavior policy, its safety performance is at least guaranteed to match that of the behavior policy underlying the dataset.

To theoretically ground our algorithm, we analyze its convergence properties in the tabular setting, where the state and action spaces are finite (i.e., $|\mathcal{S}|, |\mathcal{A}| < \infty$). Under softmax parameterization, we derive convergence guarantees when optimizing the value function using the natural policy gradient (NPG) algorithm.

**Definition 3.1** (Softmax Parameterization). *Under the tabular MDP setting, the policy follows a softmax parameterization, where the policy $\pi$ is parameterized by $\theta : \mathcal{S} \times \mathcal{A} \to \mathbb{R}$. The policy is expressed as $\pi_\theta(a|s) = \frac{\exp(\theta(s,a))}{\sum_{a' \in \mathcal{A}} \exp(\theta(s, a'))}$.*

---

**Algorithm 1** DRCORL

---
1: **Input:** Dataset $\mathcal{D}^\mu$, slack bounds $h^+, h^-$
2: Pretrain diffusion model $\epsilon_\psi$
3: *// Behavior Cloning*
4: Pretrain $Q_r^\pi, Q_c^\pi$
5: *// Pretrain critic*
6: **for** each gradient step $t$ **do**
7:     Sample mini-batch $\mathcal{B}$
8:     Update critics for reward and cost
9:     $g \leftarrow \text{SafeAdaptation}(\dots)$
10:     Update $\theta \leftarrow \theta + \eta g$
11: **end for**

---

We use natural policy gradient method [Kakade and Langford, 2002] to update policy, where the policy parameter $\theta$ is updated as $\theta \leftarrow \theta + \eta(\mathcal{F}_\rho^\theta)^\dagger \nabla_\theta V^\pi(\rho)$ at each iteration, where $\mathcal{F}_\rho^\theta = \mathbb{E}_{s \sim \rho, a \sim \pi} [\nabla_\theta \log \pi_\theta(a|s) \nabla_\theta \log \pi_\theta(a|s)^\top]$ is the Fisher information matrix, and the $\dagger$ operator is the Moore-Penrose inverse.

Since the score network is trained on the empirical offline distribution, there exists a potential shift between the dataset and the evolving policy distribution. The regret bounds in Theorem 3.1 explicitly account for this effect through the term $\epsilon_{\text{offline}}$, which measures the approximation gap between these distributions. In practical implementation, this term remains small when the dataset adequately covers the behavioral policy, ensuring that safety guarantees degrade gracefully even under mild mismatch.

**Theorem 3.1.** *Let $\tilde{\pi}$ be the weighted policy obtained after $T$ iterations of Algorithm 1 with proper step-sizes. Suppose that the offline dataset has a size of $|\mathcal{D}^\mu| = \mathcal{O}\left(\frac{C \ln(|\mathcal{F}|/\delta)}{\epsilon_{\text{offline}}(1-\gamma)^4}\right)$ for some $\delta \in (0, 1)$, where $\mathcal{F}$ represents the critic function family. Then, with probability $\geq 1 - \delta$, the optimality gap and constraint violation satisfy that*

$$V_r^{\pi_{\theta^*}}(\rho) - \mathbb{E}[V_r^{\tilde{\pi}}(\rho)] \leq \mathcal{O}(\epsilon_{\text{offline}}) + \mathcal{O}\left(\sqrt{\frac{|\mathcal{S}||\mathcal{A}|}{(1-\gamma)^3 T}}\right), \tag{14}$$

$$\mathbb{E}[V_c^{\tilde{\pi}}(\rho)] - b \leq \mathcal{O}(\epsilon_{\text{offline}}) + \mathcal{O}\left(\sqrt{\frac{|\mathcal{S}||\mathcal{A}|}{(1-\gamma)^3 T}}\right). \tag{15}$$

*where $\epsilon_{offline}$ denotes the approximation error of policy evaluation induced by the offline dataset $\mathcal{D}^\mu$.*

For simplicity, we drop the reverse KL term to ensure the policy's closed-form update under NPG in our proof. The SafeAdaptation step in Algorithm 1 is specified in algorithm 2 given in Appendix B. We specify the definition of the weighted policy $\tilde{\pi}$ in Eq. (41) given in Appendix A.2. We can interpret the theorem as follows: by selecting appropriate slack bounds $h^+$ and $h^-$, the policy optimization process will, over time, primarily remain in the reward optimization and gradient manipulation stages. As a result, the cost violation can be effectively upper bounded. Simultaneously, by ensuring that the policy is updated using the reward critic's gradients for the majority of iterations, we can guarantee that the accumulated reward of the weighted policy closely approximates that of the optimal policy.

# 4    Practical Algorithm

In this section, we present the detailed procedure for the implementation of our main algorithm *Diffusion-Regularized Constrained Offline Safe Reinforcement Learning* (DRCORL), as outlined in Algorithm 1, where we provide the SafeAdaptation step outlined in Algorithm 2 given in Appendix B. This includes both the pretraining stage and the policy extraction stage.

## 4.1    Pretraining the Diffusion Policy and Critics

In the pretraining stage, we first use the offline dataset $\mathcal{D}^\mu$ to pretrain the diffusion policy $\mu_\psi(a|s)$ to simply imitate the behavioral policy by minimizing the loss in Eq. (5). Then we also pretrain the critic functions $Q_r^\pi$ and $Q_c^\pi$, but we pretrain the reward critic with Implicit Q-Learning (IQL) Kostrikov et al. [2021] and pretrain the cost critic with TD learning with pessimistic estimation. We utilize IQL to train the reward critic function by maintaining two Q-functions $\left(Q_{r_1}^\pi, Q_{r_2}^\pi\right)$ and one value function $V$ as the critic for the reward. The loss function for the value function $V_r^\pi$ is defined as:

$$L_{V_r^\pi} = \mathbb{E}_{s,a\sim\mathcal{D}^\mu}[L_2^\tau(\min(Q_{r_1}^\pi(s,a), Q_{r_2}^\pi(s,a)) - V_r^\pi(s))], \tag{16}$$

where $L_2^\tau(u) = |\tau - \mathbb{I}(u < 0)|u^2$, and $\tau \in [0, 1]$ is a quantile. When $\tau = 0.5$, $L_2^\tau$ simplifies to the $L_2$ loss. When $\tau > 0.5$, $L_2^\tau$ encourages the learning of the $\tau$ quantile values of $Q$. The loss function for updating $Q_{r_i}^\pi$ is given by:

$$\mathcal{L}_{Q_{r_i}^\pi} = \mathbb{E}_{(s,a,s'\sim\mathcal{D}^\mu)}\left[\left\|r(s,a) + \gamma V_r^\pi(s') - Q_{r_i}^\pi(s,a)\right\|^2\right]. \tag{17}$$

This setup aims to minimize the error between the predicted Q-values and the target values derived from the value function $V_r^\pi$. We employ IQL to pretrain the reward critic function, thereby approximating the optimal Q-function $Q_r^*$ without incorporating safety constraints. Additionally, we pretrain the cost critics using the temporal difference method and double-Q learning Sutton [1988]. However, in the offline RL setting, we adopt a pessimistic estimation approach for the cost critic using a positive hyperparameter $\alpha$ to avoid underestimation as stated in Eq. (18), thereby preventing the learning of unsafe policies. The cost critic can be updated by solving the optimization problem:

$$\min_{\pi\in\Pi} \mathbb{E}_{s,a,s,a'\sim\pi(\cdot|s')}\left[\left\|c(s,a) + \gamma Q_c^\pi(s',a') - Q_c^\pi(s,a)\right\|^2\right] - \alpha\mathbb{E}_{s,a\sim\pi}[Q_c^\pi(s,a)]. \tag{18}$$

## 4.2    Extracting Policy

Now, we extract the policy $\pi_\theta$ from the diffusion model $\epsilon_\psi$ and the pretrained critic functions. Note that at this stage we need to optimize the reward while preventing the unsafe performance. Therefore, we define two optimization targets, one for the reward and one for the cost. The reward optimization target is defined as maximizing the critic value regularized by the KL-divergence with respect to the behavioral policy, a smaller value of the temperature $\beta$ refers to the higher conservativeness of our algorithm:

$$\max \mathbb{E}_{s,a\sim\pi_\theta}\left[Q_r^{\pi_\theta}(s,a) - \frac{1}{\beta}l(\pi_\theta, \mu_\psi|s)\right], \tag{19}$$

where $l(\pi_\theta, \mu_\psi|s)$ denotes the KL-divergence $D_{\text{KL}}\left(\pi_\theta(\cdot|s)||\mu_\psi(\cdot|s)\right)$ for abbreviation. Similarly, we define the cost optimization target as follows. We aim to minimize the cost critic value regularized by the behavioral policy:

$$\max \mathbb{E}_{s,a\sim\pi_\theta}\left[-(Q_c^{\pi_\theta}(s,a) - l) - \frac{1}{\beta}l(\pi_\theta, \mu_\psi|s)\right]. \tag{20}$$

Using the result obtained in Section 3.2, we can obtain the gradient of Eq. (19) and Eq. (20) with respect to $\theta$ using the score function of the diffusion model $\epsilon_\psi$ and the critic function. We take the Gaussian policy family under a constant variance. For example, with $\Pi := \{\pi_\theta(a|s) = \mathcal{N}(a; m_\theta(s), \Sigma_\theta(s)\}$, we can simplify the gradient to

$$
\begin{aligned}
g_r &= \mathbb{E}_{s, a \sim \pi_\theta} \left[ \left( \nabla_a Q_r^{\pi_\theta}(s, a) + \tfrac{1}{\beta} h_\psi(s, a) \right) \nabla_\theta \pi_\theta(s) \right], \\
g_c &= \mathbb{E}_{s, a \sim \pi_\theta} \left[ \left( -\nabla_a Q_c^{\pi_\theta}(s, a) + \tfrac{1}{\beta} h_\psi(s, a) \right) \nabla_\theta \pi_\theta(s) \right].
\end{aligned}
\tag{21}
$$

where $h_\psi(s, a) = \frac{1}{\sqrt{\bar{\beta}_t}} \epsilon_\psi(a_t, t|s)|_{t \to 0}$. In Eq. (21), $a_t$ denotes the action with perturbed noise, $a_t = \sqrt{\bar{\alpha}_t} a + \sqrt{\bar{\beta}_t} \epsilon_t$, $\epsilon_t \sim \mathcal{N}(0, I)$, and the notation $\nabla_\theta \pi_\theta(s)$ denotes the gradient of the action given state $s$ with respect to $\theta$. By assuming $a = m_\theta(s) + \Sigma_\theta^{1/2}(s) z$ where $z \sim \mathcal{N}(0, I)$, we obtain that $\nabla_\theta \pi_\theta(s) = \nabla_\theta m_\theta(s) + \nabla_\theta \Sigma_\theta^{1/2}(s) z$. Our problem involves two competing objectives: maximizing reward (Eq. 10) and minimizing cost under safety violations (Eq. 20). To reconcile these goals, we adopt the gradient manipulation method from Gu et al. [2024a], originally proposed for online safe RL, as detailed in Algorithm 2. This method introduces slack variables $h^-$ and $h^+$. When $V_c^\pi(\rho) \leq l - h^-$, the policy is considered safe, and reward maximization is prioritized. If $V_c^\pi(\rho) \geq l + h^+$, we instead focus on cost minimization. Within the transition band $l - h^- \leq V_c^\pi(\rho) \leq l + h^+$, we apply gradient manipulation (Eq. 12) to balance the two objectives (Eqs. 19 and 20).

A key challenge is accurate safety assessment in the offline setting, where off-policy data may lead to cost underestimation. Ideally, the estimated critic satisfies $Q_c^\pi = \hat{Q}_c^\pi + \epsilon$, with $\epsilon$ zero-mean. However, under $\mathbb{E}_{s, a \sim \pi}[\min \hat{Q}_c^\pi(s, a)] \leq \min \mathbb{E}_{s, a \sim \pi}[Q_c^\pi(s, a)]$, the error becomes biased, leading to underestimated cost values Thrun and Schwartz [2014]. Temporal difference learning can amplify this bias. While conservative Q-learning mitigates underestimation, it often yields sub-optimal policies. Crucially, unlike reward critics, where relative ranking suffices, underestimating cost critics can produce unsafe policies. To address this, we adopt a conservative evaluation approach inspired by the UCB (Upper Confidence Bound) technique Hao et al. [2019]. Specifically, we train an ensemble of cost critics $Q_c^{\pi, i}$ for $i = 1, \ldots, E$ and define the UCB estimator as $Q_c^{\pi, \text{UCB}}(s, a) = \bar{Q}_c^\pi(s, a) + k \cdot \text{Std}_{i \in [E]}(Q_c^{\pi, i}(s, a))$, where $\bar{Q}_c^\pi(s, a)$ is the ensemble mean and $k$ controls the confidence level. We then compute $Q_c^{\pi, \text{UCB}}(\rho) = \mathbb{E}_{s \sim \rho, a \sim \pi}[Q_c^{\pi, \text{UCB}}(s, a)]$ and compare it against the cost budget to determine whether to prioritize reward or cost. This full procedure is outlined in Algorithm 2.

## 5 Experiments

### 5.1 Performance On DSRL Benchmarks

**Environments.** We evaluate our method on the offline safe RL benchmark DSRL Liu et al. [2023a]. We conduct extensive experiments on Safety-Gymnasium Ray et al. [2019] and Bullet-Safety-Gym Gronauer [2022]. We evaluate the score of different methods using the normalized returns and normalized costs. The normalized returns and costs are defined as $R_{\text{normalized}} = (R_\pi - R_{\min})/(R_{\max} - R_{\min})$, $C_{\text{normalized}} = (C_\pi - C_{\min})/(l + \epsilon)$. $\epsilon$ is a regularizer for the case when $l = 0$, and we set $\epsilon = 0.1$. The reward $R_\pi$ is the accumulated return collected within an episode $R_\pi = \sum_{t=1}^T r_t$. Similarly, $C_\pi = \sum_{t=1}^T c_t$ is the accumulated cost collected within an episode. $R_{\max}$, $R_{\min}$ are the maximum and minimum accumulated returns of the offline dataset within an episode. We normalize the accumulated cost and return to better analyze the results. If the normalized cost is below 1.0, we can consider this policy as safe.

**Baseline Algorithms.** We compare the performance of our algorithm against existing offline safe reinforcement learning (RL) algorithms under the soft constraints setting. The following six baseline algorithms are considered: 1) *BC-Safe* (Behavioral Cloning): cloning the safe trajectories within the offline dataset. 2) *BCQ-Lag* Fujimoto et al. [2019]: this approach extends behavioral cloning with Q-learning, framing safe reinforcement learning as a constrained optimization problem. A Lagrangian multiplier Stooke et al. [2020] is used to balance the reward maximization objective with the cost constraints. 3) *BEARL* Kumar et al. [2019]: an extension of BEAR that incorporates a Lagrangian multiplier Stooke et al. [2020] to control cost constraints, enabling safe policy learning. 4) *CDT (Constrained Decision Transformer)* Liu et al. [2023b]: an adaptation of the Decision Transformer

Table 1: Normalized return (↑: higher is better) and cost (↓: lower is better; threshold at 1) for each policy across tasks. Results are averaged over three cost limit scales, 20 evaluation episodes and 5 random seeds. Policies with normalized cost ≤ 1 (safe) are bolded; among these, the highest-reward safe policy per task is highlighted in blue. Policies with cost > 1 (unsafe) are shown in gray.

| | BC-Safe | | BEARL | | BCQ-Lag | | CPQ | | COptiDICE | | CDT | | CAPS | | CCAC | | Ours | |
|---|---|---|---|---|---|---|---|---|---|---|---|---|---|---|---|---|---|---|
| Task | reward ↑ | cost ↓ | reward ↑ | cost ↓ | reward ↑ | cost ↓ | reward ↑ | cost ↓ | reward ↑ | cost ↓ | reward ↑ | cost ↓ | reward ↑ | cost ↓ | reward ↑ | cost ↓ | reward ↑ | cost ↓ |
| CarGoal1 | 0.38 | 0.46 | 0.71 | 4.45 | 0.46 | 3.27 | 0.68 | 3.73 | 0.48 | 2.31 | 0.65 | 3.75 | 0.40 | 1.35 | 0.84 | 6.52 | 0.91 | 0.00 |
| CarGoal2 | 0.25 | 0.82 | 0.46 | 10.98 | 0.29 | 3.46 | 0.27 | 28.24 | 0.18 | 2.28 | 0.03 | 0.00 | 0.12 | 2.20 | 0.96 | 6.97 | 0.79 | 0.60 |
| PointButton1 | 0.17 | 1.37 | 0.35 | 6.71 | 0.17 | 4.00 | 0.56 | 11.63 | 0.09 | 3.55 | 0.62 | 13.05 | 0.16 | 3.66 | 0.71 | 4.27 | 0.81 | 0.40 |
| PointCircle1 | 0.88 | 3.79 | -0.33 | 17.84 | 0.45 | 8.13 | 0.23 | 6.77 | 0.85 | 18.30 | 0.51 | 0.00 | 0.50 | 0.14 | 0.62 | 7.58 | 0.53 | 0.40 |
| PointGoal1 | 0.53 | 0.88 | 0.76 | 3.46 | 0.59 | 5.06 | 0.41 | 0.69 | 0.52 | 5.26 | 0.68 | 4.35 | 0.20 | 0.53 | 0.77 | 5.20 | 0.88 | 0.00 |
| PointGoal2 | 0.60 | 3.15 | 0.80 | 12.33 | 0.65 | 10.80 | 0.34 | 5.10 | 0.38 | 4.62 | 0.32 | 1.45 | 0.23 | 1.91 | 0.80 | 4.88 | 0.82 | 0.00 |
| AntVel | 0.87 | 0.52 | -1.01 | 0.00 | 0.99 | 8.39 | -1.01 | 0.00 | 0.99 | 11.41 | 0.91 | 0.97 | 0.81 | 0.36 | 0.71 | 0.39 | 0.88 | 0.89 |
| HalfCheetahVel | 0.94 | 0.71 | 0.95 | 104.45 | 1.06 | 63.94 | 0.71 | 14.70 | 0.61 | 0.00 | 0.96 | 0.61 | 0.88 | 0.22 | 0.85 | 0.93 | 0.86 | 0.00 |
| HopperVel | 0.21 | 1.50 | 0.26 | 124.05 | 0.81 | 14.91 | 0.57 | 0.00 | 0.14 | 7.83 | 0.21 | 1.12 | 0.83 | 0.00 | 0.11 | 0.68 | 0.68 | 0.79 |
| Walker2dVel | 0.78 | 0.08 | 0.76 | 0.80 | 0.80 | 0.07 | 0.08 | 0.95 | 0.12 | 2.34 | 0.73 | 1.95 | 0.78 | 0.00 | 0.21 | 0.26 | 0.74 | 0.30 |
| CarRun | 0.97 | 0.10 | 0.49 | 7.43 | 0.95 | 0.00 | 0.92 | 0.05 | 0.93 | 0.00 | 0.99 | 1.10 | 0.98 | 0.86 | 0.93 | 0.05 | 0.99 | 0.30 |
| BallCircle | 0.55 | 0.93 | 0.90 | 5.79 | 0.68 | 3.79 | 0.64 | 0.07 | 0.47 | 4.71 | 0.68 | 2.05 | 0.56 | 0.18 | 0.73 | 0.14 | 0.78 | 0.00 |
| CarCircle | 0.64 | 2.97 | 0.73 | 1.41 | 0.62 | 2.88 | 0.70 | 0.00 | 0.47 | 5.81 | 0.73 | 2.24 | 0.57 | 0.00 | 0.72 | 0.22 | 0.68 | 0.79 |

Chen et al. [2021] for offline safe reinforcement learning, incorporating cost information into tokens to learn a safe policy. 5) *CPQ (Conservative Policy Q-Learning)* Xu et al. [2022]: this method uses conservative Q-learning to pessimistically estimate the cost critic and updates the reward critic only with safe cost values, ensuring the policy adheres to safety constraints. 6) *COptiDICE* Lee et al. [2022]: based on the DICE algorithm, this method learns a stationary distribution for the safe reinforcement learning problem and extracts the optimal policy from this distribution. 7) *CAPS* Chemingui et al. [2025]: optimizing different constraints with shared representations. 8) *CCAC* Guo et al. [2025]: safety RL algorithm matching state-action distributions and safety constraints. We present the comparison across different baselines in SafetyGym and BulletGym environments.

**Result Analysis.** Table 1 summarizes the normalized accumulated reward and cost per episode across tasks. Overall, our algorithm consistently achieves high rewards while reliably maintaining safety constraints. Notably, our method significantly outperforms baselines in tasks such as *CarGoal1*, *CarGoal2*, *PointGoal1*, *PointGoal2*, and *BallCircle*, ensuring both optimal reward and constraint satisfaction. However, it is fair to acknowledge that BCQ-Lag shows slightly superior reward performance in the *Walker2dVel* task, though such performance does not generalize reliably to other tasks, often violating safety constraints. Similarly, CDT and CAPS achieve good safety in some scenarios, such as *CarGoal2* and *AntVel*, yet underperform significantly in reward optimization in other tasks. Overall, our method demonstrates a robust balance between safety and reward optimization across diverse benchmarks, highlighting its practical applicability.

**Computational Efficiency.** We benchmark inference speeds across algorithms using 1,000 sample inputs on the HalfCheetah task. Apart from the 6 baselines above, We also compare the computation efficiency against diffusion model-based algorithms TREBI Lin et al. [2023] and FISOR Zheng et al. [2024]. While baseline methods like BCQ and CPQ achieve the fastest inference due to their lightweight MLP-based actors, they compromise safety or reward performance. In contrast, our algorithm strictly adheres to cost constraints without sacrificing reward quality. Compared to diffusion or transformer-based approaches our method are superior in inference speed, narrowing the gap between expressive generative models and efficient MLP architectures. The result is in Figure 2 (a).

## 5.2 Ablation Study

**Impact of Different Cost Limits.** We evaluate our algorithm's performance under varying cost limits $l = 10, 20$, and $30$, analyzing the learned policies' behavior for each budget setting. The ablation results are presented in Figure 2 (b). Across all cost limit choices, our algorithm consistently achieves zero violations of the safety constraints, demonstrating its robustness to varying cost thresholds. This highlights the adaptability and reliability of our approach in maintaining safety compliance. While COptiDICE also strictly adheres to the cost limits, our method consistently outperforms it in terms of normalized return, demonstrating its superior ability to balance safety and reward optimization.

**Temperature Parameter Setting.** We compare three diffusion-temperature schedules: (i) constant, $\beta_t = 0.02$; (ii) square-root growth, $\beta_t = 0.01\sqrt{t}$; and (iii) linear growth, $\beta_t = 0.01\,t + 0.04$, held fixed within each epoch. As shown in Figure 3 (Appendix C), all three schemes yield similar performance. Additional temperature ablations are reported in Appendix C.1.

**Choice of Slack Variable.** We introduce slack bounds so that reward maximization applies when $V_{\mathrm{norm}} \leq 1 - h^-$ and cost minimization when $V_{\mathrm{norm}} \geq 1 + h^+$. Both $h^-$ and $h^+$ are initialized to 0.2 and linearly decayed to zero; we ablate over initial values $h \in \{0.1, 0.3, 0.5, 0.7\}$. Figure 4 (Appendix C) illustrates these results, with further slack-value studies in Appendix C.1.

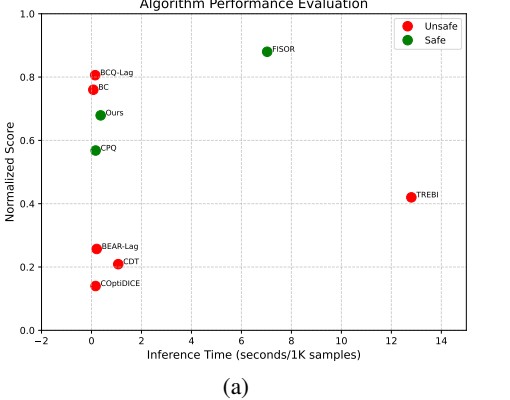
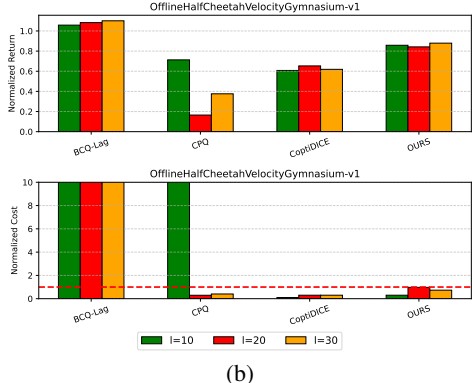

(a)                                (b)

Figure 2: (a) Computational Efficiency vs. Performance Trade-off. Normalized score ($y$-axis, combining reward and safety metrics) versus inference time for generating 1,000 actions ($x$-axis). (b) Normalized return and cost under varying cost limits ($l = 10, 20, 30$). Since the cost is normalized relative to the corresponding cost limits, the safety threshold is consistently 1.0. The dashed line represents the safety boundary.

# 6  Related Work

**Diffusion Models in Offline RL.** Diffusion models' powerful generative capacity for complex data has made them increasingly popular for modeling diverse offline datasets, valuable either for planning Janner et al. [2022], He et al. [2024], Ajay et al. [2022] or expressive policy modeling Chi et al. [2023], Wang et al. [2022], Fang et al. [2024], Hansen-Estruch et al. [2023], Chen et al. [2023], Lu et al. [2023]. Their main drawback is inherently slow sampling due to many reverse-diffusion steps. To accelerate inference, methods like DDIM Song et al. [2020a] reduce steps via subsequence sampling, while DPM-Solver Lu et al. [2022] uses an optimized ODE solver to reach ∼10 steps. Useful extensions to safety-critical settings include the cost-constrained diffuser framework of Lin et al. [2023], SafeDiffuser Xiao et al. [2023] and OASIS's conditional diffusion for dataset synthesis and careful distribution shaping Yao et al. [2024].

**Online Safe RL.** Safe RL in online environments has been extensively surveyed in Gu et al. [2024c]. A common approach is constrained policy optimization—e.g., CPO Achiam et al. [2017] integrates TRPO Schulman [2015] for robust updates, and CRPO Xu et al. [2021] obviates complex dual-variable tuning. Primal–dual Lagrangian methods dynamically adjust multipliers on the fly to enforce safety criteria [Chow et al., 2018, Calian et al., 2020, Ding et al., 2020, Ying et al., 2022, Zhou et al., 2023], and more recent algorithms aim to better balance cost and reward via gradient manipulation Gu et al. [2024a] or refined trust-region formulations Kim et al. [2023, 2024]. However, these methods depend on extensive environment interaction and accurate critic estimates, thus limiting their practicality under stringent real-world safety and data collection cost constraints.

**Safe Offline RL.** Considering safety specifically in the offline learning setting, Xu et al. [2022], Guan et al. [2023] penalized OOD and unsafe actions identified in the dataset. Diffusion-based safe policy learning was adapted by Lin et al. [2023] for cost-constrained scenarios, and decision-transformer architectures have been applied to safety considerations via Liu et al. [2023b]. Energy-based diffusion policies can enforce hard constraints, for example, through weighted regression [Lu et al., 2023, Zheng et al., 2024, Koirala et al., 2024]. More recent approaches learn by modelling conditional

sequences as in Gong et al. [2025], Zhang et al. [2023] or by strategically sharing representations across constraints for improved adaptation Chemingui et al. [2025].

# 7 Conclusion and Limitations

We present DRCORL, a novel approach for offline constrained reinforcement learning that learns safe, high-performance policies from fixed datasets. DRCORL employs diffusion models to faithfully capture offline behavioral policies and distills them into simplified policies for fast inference. To balance reward maximization and constraint satisfaction, we utilize a gradient manipulation strategy that adapts dynamically without extensive hyperparameter tuning. Extensive experiments across standard safety benchmarks show that DRCORL consistently outperforms existing offline safe RL methods, achieving superior rewards while satisfying safety constraints.

We also identify several limitations offering avenues for future work. First, DRCORL's pretraining stage comprising diffusion-model training and critic estimation, though producing lightweight policy at inference and avoids unsafe online exploration, incurs additional computational overhead. Future work should aim to optimize and parallelize the pretraining phase to further reduce its cost. Second, guaranteeing zero constraint violations remains challenging in offline RL, particularly when dataset quality and coverage vary. Developing methods with greater robustness to imperfect data and tighter safety guarantees is a promising direction. Despite these challenges, our contributions lay a strong foundation for advancing both generalization and safety in constrained offline reinforcement learning. Although DRCORL inherits the stability of diffusion modeling, its effectiveness still depends on the representational quality of the offline dataset. In extremely sparse or biased datasets, the diffusion model may under-represent safe modes. Future directions include integrating conservative density estimation or uncertainty-aware diffusion objectives to enhance robustness in low-coverage regimes. More generally, we view diffusion regularization as a flexible template: alternative generative priors such as VAEs or normalizing flows could replace the diffusion estimator under the same theoretical framework, potentially trading expressivity for computational efficiency.

Finally, we hope this work provides a unifying perspective linking generative policy modeling and safety-constrained optimization, paving the way for scalable and verifiable offline safe RL.

# 8 Acknowledgment

This material is based upon work supported in part by the U.S. Army Research Laboratory and the U.S. Army Research Office under grant number W911NF2010219. It was also supported by the Office of Naval research under grant number N000142412673, as well as NSF.

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

# A  Proofs of Theoretical Results

In this section, we provide the proof of the theoretical results in the main body. We make this assumption throughout the proof of Proposition 3.1 and Theorem 3.1.

**Assumption A.1.** *During the training process, we assume the policy $\pi_\theta$ resides in the region $\Pi_\Theta$ within the policy space. For all $\pi \in \Pi_\Theta$, $s \in \mathcal{S}$, and $a \in \mathcal{A}$, the following holds:*

$$\left| \log \frac{\pi(a|s)}{\pi_b(a|s)} \right| \leq \epsilon_{dist}. \tag{22}$$

*Equivalently, it means that for all $s \in \mathcal{S}$ and $a \in \mathcal{A}$:*

$$\exp(-\epsilon_{dist}) \leq \frac{\pi(a|s)}{\pi_b(a|s)} \leq \exp(\epsilon_{dist}). \tag{23}$$

## A.1  Proof of Proposition 3.1

*Proof.* Using the performance difference lemma [Agarwal et al., 2021, Lemma 2], we can write the difference of the value functions under policies $\tilde{\pi}$ and $\pi_b$ as

$$V_c^{\pi_b}(\rho) - V_c^{\tilde{\pi}}(\rho) = \frac{1}{1-\gamma} \mathbb{E}_{s \sim d_\rho^{\pi_b}} \mathbb{E}_{a \sim \pi_b(\cdot|s)}[A_c^{\tilde{\pi}}(s,a)], \tag{24}$$

where $d_\rho^{\pi_b}$ is the discounted stationary distribution defined as $d_\rho^{\pi_b}(s) = (1-\gamma) \sum_{t=0}^{\infty} \Pr(s_t = s | s_0 \sim \rho)$. The total variation distance (TV-distance) between two distribution is defined as $D_{\mathrm{TV}}(\tilde{\pi}(\cdot|s) || \pi_b(\cdot|s)) = \frac{1}{2} \int_\mathcal{A} |\tilde{\pi}(a|s) - \pi_b(a|s)| da$, which is proportion to the $\ell^1$-distance between the two distributions. Using the Pinsker's inequality [Csiszár and Körner, 2011], we can bound the TV-distance with the KL-divergence, i.e., for any two distributions $\mu, \nu$, we have that

$$D_{\mathrm{TV}}(\mu||\nu) \leq \sqrt{\frac{D_{\mathrm{KL}}(\mu||\nu)}{2}}. \tag{25}$$

Hence, it holds that $D_{\mathrm{TV}}(\tilde{\pi}||\pi_b) \leq \sqrt{\epsilon_{\mathrm{dist}}/2}$. Then, following the same procedure as Achiam et al. [2017, Proposition 2], we can obtain that

$$V_c^{\pi_b}(\rho) - V_c^{\tilde{\pi}}(\rho) \geq \underbrace{\frac{1}{1-\gamma} \mathbb{E}_{s \sim d_\rho^{\tilde{\pi}}} \mathbb{E}_{a \sim \pi_b(\cdot|s)}[A_c^{\tilde{\pi}}(s,a)]}_{\mathrm{I}} - \underbrace{\frac{2\gamma \epsilon_{\mathrm{adv}}^b}{(1-\gamma)^2} \mathbb{E}_{s \sim d_\rho^{\tilde{\pi}}}[D_{\mathrm{TV}}(\tilde{\pi}(\cdot|s) || \pi_b(\cdot|s))]}_{\mathrm{II}}. \tag{26}$$

Using Eq. (25), we can bound term II with

$$\frac{2\gamma \epsilon_{\mathrm{adv}}^b}{(1-\gamma)^2} \mathbb{E}_{s \sim d_\rho^{\tilde{\pi}}}[D_{\mathrm{TV}}(\tilde{\pi}(\cdot|s) || \pi_b(\cdot|s))] \leq \frac{\gamma \epsilon_{\mathrm{adv}}^b \sqrt{2\epsilon_{\mathrm{dist}}}}{(1-\gamma)^2}. \tag{27}$$

As for term I, we can bound it with

$$\begin{aligned}
\left| \frac{1}{1-\gamma} \mathbb{E}_{s \sim d_\rho^{\tilde{\pi}}} \mathbb{E}_{a \sim \pi_b(\cdot|s)}[A_c^{\tilde{\pi}}(s,a)] \right| &\leq \frac{1}{1-\gamma} \mathbb{E}_{s \sim d_\rho^{\tilde{\pi}}} \left| \int_\mathcal{A} \pi_b(a|s) Q^{\tilde{\pi}}(s,a) - \tilde{\pi}(a|s) Q^{\tilde{\pi}}(s,a) da \right| \\
&\leq \frac{1}{1-\gamma} \mathbb{E}_{s \sim d_\rho^{\tilde{\pi}}} \int_\mathcal{A} |\pi_b(a|s) - \tilde{\pi}(a|s)| \underbrace{Q^{\tilde{\pi}}(s,a)}_{\in [0, \frac{c_{\max}}{1-\gamma}]} da \\
&\leq \frac{c_{\max}}{(1-\gamma)^2} \mathbb{E}_{s \sim d_\rho^{\tilde{\pi}}} \int_\mathcal{A} |\tilde{\pi}(a|s) - \pi_b(a|s)| da \\
&\leq \frac{2 c_{\max}}{(1-\gamma)^2} \mathbb{E}_{s \sim d_\rho^{\tilde{\pi}}}[D_{\mathrm{TV}}(\tilde{\pi}(\cdot|s) || \pi_b(\cdot|s))] \\
&\leq \frac{c_{\max} \sqrt{2\epsilon_{\mathrm{dist}}}}{(1-\gamma)^2}.
\end{aligned} \tag{28}$$

Combining Eq. (28) with Eq. (27), we can finally obtain that

$$V_c^{\pi_b}(\rho) - V_c^{\tilde{\pi}}(\rho) = \text{I} - \text{II} \geq -\frac{c_{\max}\sqrt{2\epsilon_{\text{dist}}}}{(1-\gamma)^2} - \frac{\gamma\epsilon_{\text{adv}}^b\sqrt{2\epsilon_{\text{dist}}}}{(1-\gamma)^2} = -\frac{(c_{\max} + \gamma\epsilon_{\text{adv}}^b)\sqrt{2\epsilon_{\text{dist}}}}{(1-\gamma)^2}, \quad (29)$$

which concludes the proof of Proposition 3.1. $\qquad\square$

## A.2 Proof of Theorem 3.1

Our proof is based on theorem 1 in Gu et al. [2024b] which is considering a online safe reinforcement learning setting. Before presenting the proof, we first introduce some notations and concepts that will be used throughout this section. We index the iterations of Algorithm 1 by $t = 1, 2, \ldots, T$.

- Let $\hat{Q}_r^t(s, a)$ and $\hat{Q}_c^t(s, a)$ denote the estimators of the critic functions $Q_r^{\pi_{\theta_t}}(s, a)$ and $Q_c^{\pi_{\theta_t}}(s, a)$, respectively, at the $t$-th iteration under the policy $\pi_{\theta_t}$.

- Denote the gradient at step $t$ for the reward optimization as $g_r^t$ and the gradient at step $t$ for the cost optimization as $g_c^t$.

- Let $\eta$ represent the learning rate for the NPG algorithm.

- We categorize the iterations of Algorithm 1 into four cases based on the optimization scenarios:

  1. **Safe Policy Region**, i.e., when $V_c^{\pi_{\theta_t}} \leq l - h^-$. We denote the set of iteration indices corresponding to this case as $S_{\text{safe}}$.
  2. **Unsafe Region**, i.e., when $V_c^{\pi_{\theta_t}} \geq l + h^+$. We denote the set of iteration indices corresponding to this case as $S_{\text{unsafe}}$.
  3. **Gradient Manipulation - Aligned Gradients**, i.e., when the cost function is close to the cost limit threshold and the angle between the updated gradients is less than $90°$. We denote the corresponding set of iteration indices as $S_{\text{align}}$.
  4. **Gradient Manipulation - Conflicting Gradients**, i.e., when the cost function is close to the cost limit threshold and the angle between the updated gradients is greater than $90°$. We denote the corresponding set of iteration indices as $S_{\text{conflict}}$.

  For every $t \in \{1, 2, \ldots, T\}$, the iteration index $t$ must belong to one of the four sets: $S_{\text{safe}}$, $S_{\text{unsafe}}$, $S_{\text{align}}$, or $S_{\text{conflict}}$.

- Assume the reward function $r : \mathcal{S} \times \mathcal{A} \to [0, M]$ and the cost function $c : \mathcal{S} \times \mathcal{A} \to [0, M]$ are non-negative and bounded by $M$. This is a standard assumption in the tabular setting.

- We define the Bellman operator $\mathcal{T}$ for policy evaluation (applicable to both cost and reward functions) as:

$$\mathcal{T}f(s, a) = r(s, a) + \gamma\mathbb{E}_{s' \sim P(\cdot|s,a)}\left[\max_{a' \in \mathcal{A}} f(s', a')\right]. \quad (30)$$

*Proof.* Under the softmax parameterization, the natural policy gradient update [Kakade and Langford, 2002] can be expressed as

$$\pi_{\theta_{t+1}}(a|s) = \pi_{\theta_t}(a|s)\frac{\exp\left(\eta Q^{\pi_{\theta_t}}(s, a)/(1-\gamma)\right)}{Z_t(s, a)}, \quad (31)$$

where the normalization constant $Z_t(s)$ is defined as:

$$Z_t(s) = \sum_{a \in \mathcal{A}} \pi_{\theta_t}(a|s)\exp\left(\frac{\eta Q^{\pi_{\theta_t}}(s, a)}{1-\gamma}\right). \quad (32)$$

Since with the reverse KL term we no longer have the close form softmax update under natural policy gradient algorithm. For simplicity we drop the KL term in the theoretical proof here as under assumption A.1 after the pretrain stage the KL term no longer dominate the loss function, and we mainly focus on the proof on the balance between reward and cost.

We admit that omitting the reverse KL divergence term simplifies the derivation of a closed-form NPG update, yet the core insights from this analysis are expected to remain relevant to the behavior of the full DRCORL algorithm. This is mainly because:

(i) The initial pretraining of the diffusion model (Section 3.1) and its subsequent use as a regularizer (Section 3.2) aim to ensure the learned policy $\pi_\theta$ primarily operates within a region close to the behavioral policy $\pi_b$. Assumption A.1 formalizes this by bounding the KL divergence.

(ii) In this regularized regime, the KL term primarily acts to constrain exploration and prevent significant deviation from the data distribution. The fundamental trade-offs and convergence dynamics related to balancing expected rewards and costs are still governed by the NPG updates on these value estimations. Thus, the analysis, while idealized, sheds light on the core learning dynamics concerning reward maximization under cost constraints.

**Assumption A.2.** *Given an offline dataset $\mathcal{D}^\mu = \{(s_i, a_i, s_i', r_i, c_i)\}_{i=1}^N$ of size $|\mathcal{D}^\mu| = N$, let the value function class be $\mathcal{F}$ and define the model class as $\mathcal{G} = \{\mathcal{T}f | f \in \mathcal{F}\}$. We assume:*

- **Realizability**: *The critic function $Q^*$, learned by optimizing Eq. (16), Eq. (17), and Eq. (18), belongs to the function class $\mathcal{F}$, and $\mathcal{T}Q^*$ resides in the model class $\mathcal{G}$. Moreover, we assume $\mathcal{G} = \mathcal{F}$.*

- **Dataset Diversity**: *The offline dataset is diverse enough to ensure accurate offline policy evaluation. Specifically, we assume that:*

$$N = \mathcal{O}\left(\frac{C\ln(|\mathcal{F}|/\delta)}{\epsilon_{\text{offline}}(1-\gamma)^4}\right), \tag{33}$$

*where $\epsilon_{\text{offline}}$ is the desired accuracy, and $\delta$ is the failure probability.*

By invoking Chen and Jiang [2019, Theorem 3], we can show that under Assumption A.2, with probability at least $1 - \delta$, the following bounds hold:

$$\|\hat{Q}_r^\pi - Q_r^{\pi,*}\| \le \epsilon_{\text{offline}}, \quad \|\hat{Q}_c^\pi - Q_c^{\pi,*}\| \le \epsilon_{\text{offline}}. \tag{34}$$

Then, by Gu et al. [2024b, Lemma A.2], we can show that the policy update with gradient manipulation satisfies that:

- For all $t \in S_{\text{safe}}$, the bound on the reward function is given by:

$$V_r^{\pi_{\theta*}}(\rho) - V_r^{\pi_{\theta_t}}(\rho) \le \frac{1}{\eta}\mathbb{E}_{s \sim d_\rho^{\pi_{\theta*}}}\left[D_{\text{KL}}(\pi_{\theta*}(\cdot|s)\|\pi_{\theta_t}(\cdot|s)) - D_{\text{KL}}(\pi_{\theta*}(\cdot|s)\|\pi_{\theta_{t+1}}(\cdot|s))\right]$$
$$+ \frac{2\eta|\mathcal{S}||\mathcal{A}|M^2}{(1-\gamma)^3} + \frac{3(1+\eta M)}{(1-\gamma)^2}\|Q_r^{\pi_{\theta_t}} - \hat{Q}_r^{\pi_{\theta_t}}\|_2. \tag{35}$$

- Similarly, for all $t \in S_{\text{unsafe}}$, the bound on the cost function is:

$$V_c^{\pi_{\theta*}}(\rho) - V_c^{\pi_{\theta_t}}(\rho) \le \frac{1}{\eta}\mathbb{E}_{s \sim d_\rho^{\pi_{\theta*}}}\left[D_{\text{KL}}(\pi_{\theta*}(\cdot|s)\|\pi_{\theta_t}(\cdot|s)) - D_{\text{KL}}(\pi_{\theta*}(\cdot|s)\|\pi_{\theta_{t+1}}(\cdot|s))\right]$$
$$+ \frac{2\eta|\mathcal{S}||\mathcal{A}|M^2}{(1-\gamma)^3} + \frac{3(1+\eta M)}{(1-\gamma)^2}\|Q_c^{\pi_{\theta_t}} - \hat{Q}_c^{\pi_{\theta_t}}\|_2. \tag{36}$$

- For $t \in S_{\text{align}}$, we have the combined bound for reward and cost as:

$$\frac{1}{2}\left(V_r^{\pi_{\theta*}}(\rho) - V_r^{\pi_{\theta_t}}(\rho)\right) + \frac{1}{2}\left(V_c^{\pi_{\theta*}}(\rho) - V_c^{\pi_{\theta_t}}(\rho)\right)$$
$$\le \frac{1}{\eta}\mathbb{E}_{s \sim d_\rho^{\pi_{\theta*}}}\left(D_{\text{KL}}(\pi_{\theta*}(\cdot|s)\|\pi_{\theta_t}(\cdot|s)) - D_{\text{KL}}(\pi_{\theta*}(\cdot|s)\|\pi_{\theta_{t+1}}(\cdot|s))\right) + \frac{2\eta M^2|\mathcal{S}||\mathcal{A}|}{(1-\gamma)^3} \tag{37}$$
$$+ \frac{3(1+\eta M)}{(1-\gamma)^2}\left[\frac{1}{2}\left\|Q_r^{\pi_{\theta_t}}(s,a) - \hat{Q}_r^{\pi_{\theta_t}}(s,a)\right\|_2 + \frac{1}{2}\left\|Q_c^{\pi_{\theta_t}}(s,a) - \hat{Q}_c^{\pi_{\theta_t}}(s,a)\right\|_2\right].$$

- Finally for $t \in S_{\text{conflict}}$, it holds that

$$\left(\frac{1}{2} - \frac{\langle g_r^t, g_c^t \rangle}{2\|g_r^t\|^2}\right)\left(V_r^{\pi_{\theta^*}}(\rho) - V_r^{\pi_{\theta_t}}(\rho)\right) + \left(\frac{1}{2} - \frac{\langle g_c^t, g_r^t \rangle}{2\|g_c^t\|^2}\right)\left(V_c^{\pi_{\theta^*}}(\rho) - V_c^{\pi_{\theta_t}}(\rho)\right)$$

$$\leq \frac{1}{\eta}\mathbb{E}_{s \sim d_\rho^{\pi_{\theta^*}}}\left(D_{\text{KL}}(\pi_{\theta^*}(\cdot|s)\|\pi_{\theta_t}(\cdot|s)) - D_{\text{KL}}(\pi_{\theta^*}(\cdot|s)\|\pi_{\theta_{t+1}}(\cdot|s))\right)$$

$$+ \frac{2\eta M^2 \left(1 - \frac{\langle g_r^t, g_c^t \rangle}{2\|g_r^t\|^2} - \frac{\langle g_r^t, g_c^t \rangle}{2\|g_c^t\|^2}\right)|\mathcal{S}||\mathcal{A}|}{(1-\gamma)^3} \tag{38}$$

$$+ \frac{3(1+\eta M)}{(1-\gamma)^2}\left[\frac{1}{2}\left\|Q_r^{\pi_{\theta_t}}(s,a) - \hat{Q}_r^{\pi_{\theta_t}}(s,a)\right\|_2 + \frac{1}{2}\left\|Q_c^{\pi_{\theta_t}}(s,a) - \hat{Q}_c^{\pi_{\theta_t}}(s,a)\right\|_2\right].$$

Summing the four equations, Eq. (35), Eq. (36), Eq. (37), and Eq. (38), we obtain that

$$\sum_{t \in S_{\text{unsafe}}}\left(V_c^{\pi_{\theta^*}}(\rho) - V_c^{\pi_{\theta_t}}(\rho)\right) + \frac{1}{2}\sum_{t \in S_{\text{align}}}\left(V_c^{\pi_{\theta^*}}(\rho) - V_c^{\pi_{\theta_t}}(\rho)\right)$$

$$+ \left(\frac{1}{2} - \frac{\langle g_r^t, g_c^t \rangle}{2\|g_c^t\|^2}\right) \cdot \left(V_c^{\pi_{\theta^*}}(\rho) - V_c^{\pi_{\theta_t}}(\rho)\right) \tag{39}$$

$$\leq \frac{1}{\eta}\mathbb{E}_{s \sim d_\rho^{\pi_{\theta^*}}}D_{\text{KL}}\left(\pi_{\theta^*}(\cdot|s)\|\pi_{\theta_0}(\cdot|s)\right) + \frac{2\eta|\mathcal{S}||\mathcal{A}|M^2 T}{(1-\gamma)^3} + e_Q,$$

where $e_Q$ is the accumulated weighted critic error, defined as:

$$e_Q = \sum_{t \in S_{\text{safe}}}\frac{3(1+\eta M)}{(1-\gamma)^2}\left\|Q_r^{\pi_{\theta_t}} - \hat{Q}_r^{\pi_{\theta_t}}\right\|_2 + \sum_{t \in S_{\text{unsafe}}}\frac{3(1+\eta M)}{(1-\gamma)^2}\left\|Q_c^{\pi_{\theta_t}} - \hat{Q}_c^{\pi_{\theta_t}}\right\|_2$$

$$+ \sum_{t \in S_{\text{align}}}\frac{3(1+\eta M)}{(1-\gamma)^2}\left[\frac{1}{2}\left\|Q_r^{\pi_{\theta_t}}(s,a) - \hat{Q}_r^{\pi_{\theta_t}}(s,a)\right\|_2 + \frac{1}{2}\left\|Q_c^{\pi_{\theta_t}}(s,a) - \hat{Q}_c^{\pi_{\theta_t}}(s,a)\right\|_2\right] \tag{40}$$

$$+ \sum_{t \in S_{\text{conflict}}}\frac{3(1+\eta M)}{(1-\gamma)^2}\left[\left(\frac{1}{2} - \frac{\langle g_r^t, g_c^t \rangle}{2\|g_r^t\|^2}\right)\left\|Q_r^{\pi_{\theta_t}}(s,a) - \hat{Q}_r^{\pi_{\theta_t}}(s,a)\right\|_2\right.$$

$$\left. + \left(\frac{1}{2} - \frac{\langle g_r^t, g_c^t \rangle}{2\|g_c^t\|^2}\right)\left\|Q_c^{\pi_{\theta_t}}(s,a) - \hat{Q}_c^{\pi_{\theta_t}}(s,a)\right\|_2\right].$$

Now, we need to upper bound the weighted critic error $e_Q$. We assume that there exists a positive constant $C$ such that $e_Q$ with $\frac{3CT(1+\eta M)}{(1-\gamma)^2}\epsilon_{\text{offline}}$ . According to Gu et al. [2024b, Lemma A.6], by choosing reasonably large values for $h^+$ and $h^-$, we can ensure that:

$$|S_{\text{unsafe}}| + |S_{\text{align}}| + |S_{\text{conflict}}| \geq T/2.$$

For example, by setting $h^+ = 2\sqrt{\frac{|\mathcal{S}||\mathcal{A}|}{(1-\gamma)^3 T}}\left(\epsilon_{\text{dist}} + 4M^2 + 6M\right)$ and $h^- = 0$, this condition holds. Now, we define the weighted policy $\tilde{\pi}$ as follows

$$\tilde{\pi}(a|s) = \frac{\sum_{t=1}^T w_t \pi_t(a|s)}{\sum_{t=1}^T w_t}, \tag{41}$$

where the policy weights are assigned based on the categories of iterations: 1. Weight $w_t = 1$ for $t \in S_{\text{safe}}$, 2. Weight $w_t = 0$ for $t \in S_{\text{unsafe}}$,

3. Weight $w_t = \frac{1}{2} - \frac{\langle g_r^t, g_c^t \rangle}{\|g_r^t\|^2}$ for $t \in S_{\text{conflict}}$. Under the weighted policy $\tilde{\pi}$, the following bound holds true for the reward value function:

$$V_r^{\pi^*} - V_r^{\tilde{\pi}} \leq \frac{1}{\frac{1}{2}\frac{T}{2}}\left(\frac{1}{\eta}\mathbb{E}_{s \sim d_\rho^{\pi_{\theta^*}}}D_{\text{KL}}(\pi_{\theta^*}(\cdot|s)\|\pi_{\theta_t}(\cdot|s)) + \frac{2\eta|\mathcal{S}||\mathcal{A}|M^2 T}{(1-\gamma)^3} + e_Q\right)$$

$$\leq \frac{4}{T}\left(\frac{1}{\eta}\epsilon_{\text{dist}} + \frac{2\eta|\mathcal{S}||\mathcal{A}|M^2 T}{(1-\gamma)^3} + \frac{3CT(1+\eta M)}{(1-\gamma)^2}\epsilon_{\text{offline}}\right). \tag{42}$$

Now, by choosing $\eta = \sqrt{\epsilon_{\text{dist}}(1-\gamma)^3/\left(2\eta|\mathcal{S}||\mathcal{A}|M^2 + 3CT\eta M(1-\gamma)\epsilon_{\text{offline}}\right)}$, we can ensure that

$$V_r^{\pi^*} - V_r^{\tilde{\pi}} \leq \frac{1}{\sqrt{T}}\sqrt{\frac{32(2\eta|\mathcal{S}||\mathcal{A}|M^2 + 3(1+\eta M)(1-\gamma)\epsilon_{\text{offline}})}{(1-\gamma)^3}} + \frac{3C\epsilon_{\text{offline}}}{(1-\gamma)^2}$$

$$= \mathcal{O}\left(\sqrt{\frac{|\mathcal{S}||\mathcal{A}|}{(1-\gamma)^3 T}}\right) + \mathcal{O}(\epsilon_{\text{offline}}). \tag{43}$$

Finally, the safety bound is given as:

$$V_c^{\tilde{\pi}}(\rho) - b$$

$$\leq \frac{4}{T}\left[\sum_{t\in S_{\text{safe}}}(V_c^{\pi_{\theta_t}}(\rho) - b) + \frac{1}{2}\sum_{t\in S_{\text{align}}}(V_c^{\pi_{\theta_t}}(\rho) - b) + \left(\frac{1}{2} - \frac{\langle g_c^t, g_r^t\rangle}{2\|g_c^t\|^2}\right)\sum_{t\in S_{\text{conflict}}}(V_c^{\pi_{\theta_t}}(\rho) - b)\right]$$

$$\leq \frac{4}{T}\left[\sum_{t\in S_{\text{safe}}}\underbrace{(\hat{V}_c^{\pi_{\theta_t}}(\rho) - b)}_{\leq -h^-} + \frac{1}{2}\sum_{t\in S_{\text{align}}}\underbrace{(\hat{V}_c^{\pi_{\theta_t}}(\rho) - b)}_{\leq h^+} + \left(\frac{1}{2} - \frac{\langle g_c^t, g_r^t\rangle}{2\|g_c^t\|^2}\right)\sum_{t\in S_{\text{conflict}}}\underbrace{(\hat{V}_c^{\pi_{\theta_t}}(\rho) - b)}_{\leq h^+}\right]$$

$$+ \frac{4}{T}\left[\sum_{t\in S_{\text{safe}}}(V_c^{\pi_{\theta_t}}(\rho) - \hat{V}_c^{\pi_{\theta_t}}(\rho)) + \sum_{t\in S_{\text{align}}}(V_c^{\pi_{\theta_t}}(\rho) - \hat{V}_c^{\pi_{\theta_t}}(\rho)) + \sum_{t\in S_{\text{conflict}}}(V_c^{\pi_{\theta_t}}(\rho) - \hat{V}_c^{\pi_{\theta_t}}(\rho))\right]$$

$$\leq 2h^+ + 2\epsilon_{\text{offline}} = \mathcal{O}(\epsilon_{\text{offline}}) + \mathcal{O}\left(\sqrt{\frac{|\mathcal{S}||\mathcal{A}|}{(1-\gamma)^3 T}}\right). \tag{44}$$

This completes the proof of Theorem 3.1. $\qquad\square$

# B    Supplemental Materials for Algorithm 1

## B.1    Clarification for the Main Body

**Inconsistency in critic update.** In the training process, we specifically adopt a conservative estimation for the cost critic, the differing update strategies for $Q_r^\pi$ and $Q_c^\pi$ are intentional and reflect their distinct roles in our framework. The cost critic $Q_c^\pi$ plays a critical role in enforcing safety, so we adopt a pessimistic update to penalize uncertain or unsafe state-action pairs—particularly in low-coverage areas—consistent with prior work on conservative critics in offline RL.

In contrast, the reward critic $Q_r^\pi$ guides policy optimization within the safe region. Applying pessimism here may lead to overly conservative behavior, reducing performance. We therefore use a standard update to maintain a balance between safety and reward. Our gradient manipulation mechanism relies on $Q_c^\pi$ to modulate the influence of each objective, making a pessimistic cost critic essential for reliable safety adaptation. We appreciate the concern about value overestimation and will explore robust reward estimation techniques in future work.

**Pretraining stage.** We pretrain the diffusion model to accurately capture the behavioral policy from the offline dataset, enabling it to serve as a stable regularizer for the Gaussian policy during training. Unlike diffusion-based planning methods, our approach uses the diffusion model's score function to penalize out-of-distribution actions, improving stability.

After pretraining, the diffusion model remains fixed. However, updating the critics during training is crucial, as they guide the gradient manipulation mechanism to balance reward maximization and safety. This separation ensures effective and stable policy optimization throughout training.

**Gradient Manipulation Stage.** the gradient manipulation method employed in our approach differs fundamentally from the naive weighted average of the two objectives. Specifically, we dynamically adjust gradient weights based on the extent of safety constraint violations, guided by slack variables. When the safety threshold is significantly violated, parameter updates are driven primarily by the gradient of the safety objective. Conversely, when the current policy strictly adheres

to safety constraints, updates are performed using only the gradient of the performance objective. In intermediate scenarios, we judiciously combine both gradients.

To address the potential instability , our approach carefully assigns weights according to the angle between gradients, as detailed in Equation (12) of the main body. Instability typically arises when the angle between gradients exceeds $90°$. Our weighting strategy effectively mitigates gradient conflicts, preventing gradient degradation and enhancing the stability and reliability of the optimization process. Additionally, our method does not require the reward and cost critics to share parameters. Instead, we temporarily freeze and save the policy parameters to implement gradient manipulation. For example, given two separate critics, $Q_r^\pi$ and $Q_c^\pi$, we first store the initial parameters of the policy $\pi_0$. We then independently compute updates using each critic. By comparing the updated policy parameters with the initial parameters, we obtain the gradient values required for manipulation. Through this scenario-based approach to parameter updates, we effectively ensure training stability.

## B.2 Safety Adaptation Step

We outline the Safe Adaptation step in Algorithm 1, specifically detailed in Algorithm 2. Our main theorem (Theorem 3.1) is derived based on the safety adaptation procedure described in Algorithm 2. Our diffusion regularization method is compatible with both CRPO Xu et al. [2021] and the gradient manipulation method Gu et al. [2024a], as implemented in Algorithm 2. Both algorithms aim to switch dynamically between reward optimization and cost optimization.

The key distinction lies in their approach to handling scenarios where the cost is close to the cost limit threshold. In Algorithm Algorithm 2 incorporates gradient manipulation in these scenarios, further stabilizing the training process by addressing conflicts between the objectives. Theorem 3.1 specifically considers the diffusion regularization algorithm equipped with the safe adaptation procedure outlined in Algorithm 2. The fundamental difference between the two algorithms lies in the criteria for switching between reward and cost optimization objectives.

---

**Algorithm 2** Gradient Manipulation Adaptation

---

**Require:** Dataset $\mathcal{D}^\mu$
**Require:** Slack variable $h^+$, $h^-$ and cost limit $l$
  **Procedure:** SafeAdaptation($\pi_\theta, \epsilon_\psi, Q_r^{\pi_\theta}, Q_c^{\pi_\theta}, h^+, h^-$)
  **if** $Q_c^{\pi,\mathrm{UCB}}(\rho) \leq l - h^-$ **then**
    Optimize reward by solving Eq. (19)
  **else if** $Q_c^{\pi,\mathrm{UCB}}(\rho) \leq l + h^+$ **then**
    Compute $g_r$ and $g_c$ with Eq. (21)
    Gradient manipulation to obtain $g$ with Eq. (12)
  **else**
    Ensure safety by solving Eq. (20)
  **end if**
  **end procedure**

---

## B.3 Hybrid Extension

**Hybrid Agents.** The assumption of the offline reinforcement learning setting can be extended by allowing the agent to incorporate partial online interactions during the training episode. This extension enables further updates to the critic function, enhancing its ability to evaluate safety conditions with greater accuracy. Since the problem remains within the scope of offline reinforcement learning, we restrict hybrid access to two specific types:

- Access to a simulator for evaluating the cost values $V_c^\pi$.
- Access to a limited number of trajectories collected by the current policy, which can be used to update the critics and policies, thereby partially mitigating the impact of distributional shift.

With the hybrid assumption, We propose two distinct approaches for evaluating costs: **Offline Evaluation** and **Online Evaluation**. For the whole offline setting, we only use the critic functions learned from the offline dataset to evaluate the cost constraints, while for the hybrid agent we allow for online trajectory access.

**Offline Evaluation**

For the hybrid agents discussed in section A, we consider two distinct forms of hybrid access to environment data.

In the fully offline setting, we estimate the cost value $V_c^\pi(\rho)$ by randomly sampling a batch of states $\mathcal{B}_s$ from the static dataset. We assume that the state distribution in the dataset $s \sim \mathcal{D}^\mu$ is sufficiently close to the target distribution $\rho$. The cost estimator is defined as:

$$\hat{V}_c^\pi(\rho) = \frac{1}{|\mathcal{B}_s|} \sum_{s \in \mathcal{B}_s} V_c^\pi(s).$$

To avoid hyperparameter tuning and additional budget constraints on the value function, we transform the value function into an estimated episodic cost. Since the value function $V_c^\pi$ can be expressed as:

$$V_c^\pi(\rho) = \frac{1}{1-\gamma} \mathbb{E}_{s \sim d_\rho^\pi, a \sim \pi(\cdot|s)} [r(s, a)],$$

we define the estimated episodic cost as $\hat{V}_c^\pi(1-\gamma)L$, where $L$ represents the episodic length.

**Online Evaluation**

- Agents are allowed to collect a limited number of trajectories to evaluate the safety of the current policy. Based on this evaluation, the agent determines whether to prioritize optimizing the reward, jointly optimizing reward and cost, or exclusively optimizing the cost. This process serves as a performance assessment of the learned policy during each episode.

- Agents can roll out a predefined number of trajectories to update the critic function in each episode. To ensure the setting remains consistent with offline reinforcement learning, the number of trajectories is strictly limited; otherwise, it would replicate a fully online reinforcement learning setting. By leveraging this partial online data, the agent mitigates overestimation errors in the critic function, thereby improving its ability to evaluate and optimize the policy effectively.

### B.4 Further Discussion on Safe Reinforcement Learning

In this section, we discuss on the difference between the hard constraint and soft constraint in safe reinforcement learning.

**Hard Constraints**. Existing works focusing on hard constraints allow no violation on safety conditions, i.e.
$$c(s_t, a_t) = 0, \forall\, t \geq 0. \tag{45}$$

**Soft Constraints.** A variety of works focus on the soft constraint setting, the safe constraint restricts the cost limit below a certain limit $l$, either within an episode or being step wise. Which can either can be episodic limit as

$$\mathbb{E}\left[\sum_{t=0}^{T} c(s_t, a_t)\right] \leq l \quad \text{or} \quad \mathbb{E}\left[\sum_{t=0}^{T} \gamma^t c(s_t, a_t)\right] \leq l, \tag{46}$$

or be the stepwise limit as
$$\mathbb{E}[c(s_t, a_t)] \leq l, a_t \sim \pi(\cdot|s_t). \tag{47}$$

Still this type of problem allow for certain degree of safety violations as in Xu et al. [2022], Lin et al. [2023], but the soft constraints also allow for broader potential policy class to further explore higher reward. We choose the soft constraint as it allows for exploration to search for higher rewards.

### B.5 Discussion on Settings

**Comparison with Offline Reinforcement Learning.** In the context of safe reinforcement learning, simply pretraining the critic before extracting the policy is insufficient for learning an optimal policy. This contrasts with approaches such as those in Chen et al. [2024, 2023], where pretraining a reward critic $Q_\phi(s, a)$ under the behavioral policy $\pi_b$ using IQL Kostrikov et al. [2021] is sufficient. These methods do not require further updates to the critic during policy extraction.

In safe reinforcement learning, however, optimization involves the reverse KL divergence term:

$$\mathbb{E}_{s,a\sim\mathcal{D}^\mu}\left[Q_\phi(s,a)-\frac{1}{\beta}D_{\mathrm{KL}}(\pi(\cdot|s)\|\mu(\cdot|s))\right], \tag{48}$$

where $\mu(\cdot|s)$ represents the behavioral policy used to collect the offline dataset. The optimal policy $\pi_\theta^*$ for Eq. (48) is given by:

$$\pi_{\theta*}(a|s)\propto\mu(a|s)\exp(\beta Q_\phi(s,a)). \tag{49}$$

Essentially, offline reinforcement learning algorithms aim to extract a policy $\pi_\theta$ that adheres to the energy-based form presented in Eq. (49). However, in the safe reinforcement learning setting, the optimal reward critic, when unconstrained, cannot be directly used for optimizing the reward objective. Therefore, it must be updated during the safety adaptation stage.

**Comparison with Constrained Reinforcement Learning.** While most safe reinforcement learning literature focuses on the online setting, the offline setting presents unique challenges for policy extraction. In the offline scenario, the agent's only access to the environment is through an offline dataset $\mathcal{D}^\mu$, which consists of transition tuples. Ideally, these transition tuples can be utilized to construct estimators for the transition dynamics $\hat{P}(s'|s,a)$, the reward function $\hat{r}(s,a)$, and the cost function $\hat{c}(s,a)$.

However, Markov decision processes (MDPs) are highly sensitive to even small changes in the reward function, requiring efficient exploitation of the offline dataset through conservative inference and penalization of out-of-distribution (OOD) actions. To address these challenges, our approach constrains the policy to remain within a defined neighborhood of the behavioral policy $\pi_b$ and adopts a pessimistic estimation of the cost critic function, effectively mitigating the risk of unsafe implementations.

## C  Details of Experiments

### C.1  Further Ablation Study

**Temperature Parameter $\beta$.** We explore two different types of $\beta$ schedules to regulate the trade-off between policy exploration and adherence to the behavioral policy.

- **Constant $\beta$ Values.** In this approach, $\beta$ is maintained as a constant throughout all epochs. A low $\beta$ value enforces conservative Q-learning by constraining the learned policy to remain close to the behavioral policy $\pi_b$. This setting prioritizes stability and minimizes divergence from the offline dataset.

- **Variant $\beta$ Values.** Here, we employ a monotonic sequence of increasing $\beta$ values over different epochs. Following the pretraining phase, the weight on optimizing the reward critic is progressively increased, or the weight on minimizing the cost critic is reduced. This dynamic adjustment encourages the policy to explore diverse strategies, allowing it to optimize returns or reduce costs effectively while gradually relaxing the conservativeness enforced during earlier stages of training.

In the initial training phase, we set $\beta$ to a low value (starting at 0.04) to ensure the policy remains close to the behavioral policy, facilitating a stable foundation and minimizing out-of-distribution actions. As training progresses, we gradually increase $\beta$, eventually reaching 1.0, to place greater emphasis on optimizing the critics. This linear scheduling allows for a smooth transition from imitation to optimization, balancing exploration and exploitation effectively. Improper tuning of $\beta$ can impact performance:

- If $\beta$ increases too rapidly: The policy may deviate prematurely from the behavioral policy, leading to instability and potential safety violations due to insufficient grounding in safe behaviors.

- If $\beta$ increases too slowly: The policy may remain overly conservative, limiting performance improvements and resulting in suboptimal reward outcomes.

Therefore, a carefully planned $\beta$ schedule is crucial to balance safety and performance. Our linear approach has demonstrated effectiveness across various tasks.

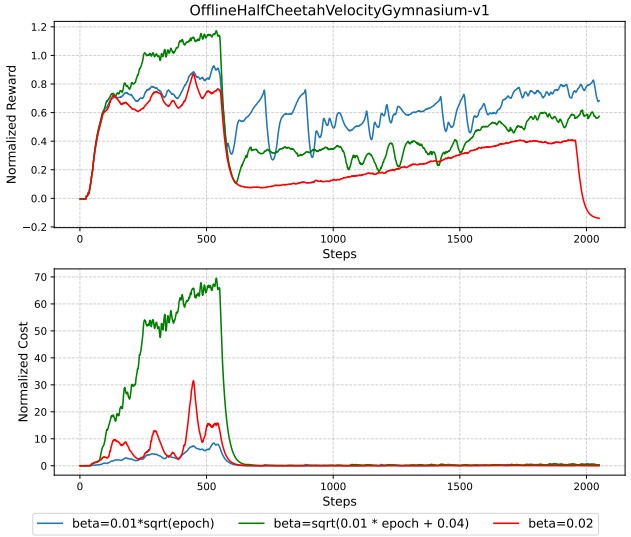

Figure 3: Training Curve Under Different Schedules. We compare the training performance of three different $\beta$ schemes, under the square root growth we have the highest normalized reward with high stability.

**Choice of Slack Variable.** We set slack bounds relative to the normalized cost so that reward maximization applies when $V_{\text{normalized}} \leq 1 - h^-$ and cost minimization triggers when $V_{\text{normalized}} \geq 1 + h^+$. During training, both $h^-$ and $h^+$ are linearly decayed from $0.2$ to zero. An ablation study on the impact of slack values is shown in Figure 4. We tested on different initial values for $h^+ = h^- = h \in \{0.1, 0.3, 0.5, 0.7\}$.

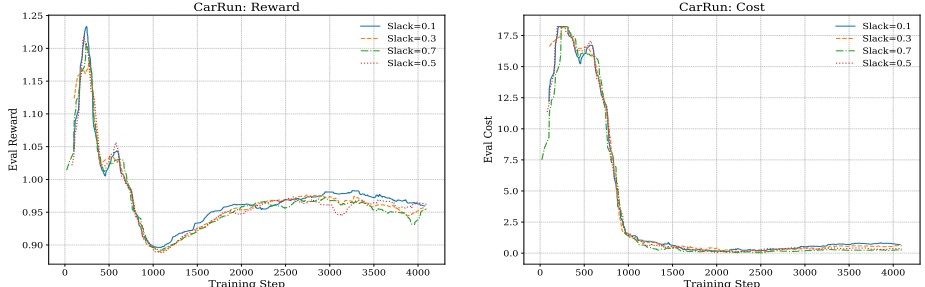

Figure 4: Slack Ablation

**Choice of cost limit.** We evaluated our algorithm on `OfflineCarRun-v0` using three cost limits ($l \in \{10, 20, 30\}$) with five random seeds each. Figure 5 presents the normalized reward and cost. The normalized reward remains consistent across different limits, and the learned policy reliably keeps the cost below the safety threshold.

We present the general hyperparameter setting in Table 2. For hyperparameters that do not apply to the corresponding algorithm, we use the back slash symbol "\" to fill the blank.

**Remark C.1.** *In Table 2, the update steps refer to the total number of gradient descent updates performed. The evaluation steps indicate the frequency of policy evaluation, measured in terms of gradient descent steps. The actor architecture (MLP) is specified as a list representing the hidden layers, where the input corresponds to the state $s$ and the output is an action $a = \pi(s)$. Similarly, the critic architecture (MLP) is represented as a list defining the hidden layers, with the input being the state-action pair and the output being a scalar value. The parameter $\tau$ represents the update rate between the target critic and the critic in double-Q learning.*

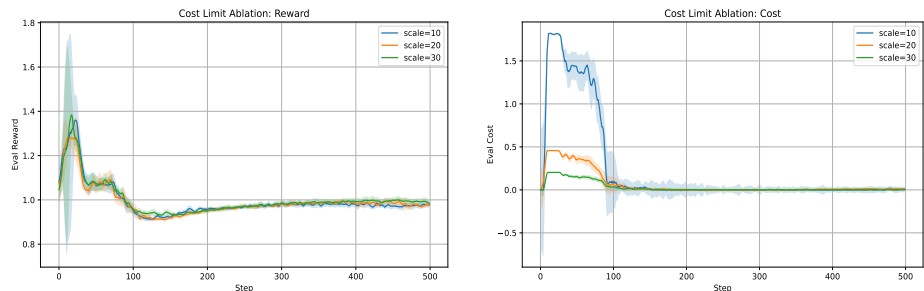

Figure 5: Cost Limit Ablation

Table 2: Summary of hyperparameter configurations for different algorithms.

| Hyperparameters | BC-Safe | BEARL | BCQ-Lag | CPQ | COptiDICE | CDT | CAPS | CCAC | Ours |
|---|---|---|---|---|---|---|---|---|---|
| Device | Cuda | Cuda | Cuda | Cuda | Cuda | Cuda | Cuda | Cuda | Cuda |
| Batch Size | 512 | 512 | 512 | 512 | 512 | 2048 | 512 | 512 | 256 |
| Update Steps | 100000 | 300000 | 100000 | 100000 | 100000 | 100000 | 100000 | 100000 | 2050 |
| Eval Steps | 2500 | 2500 | 2500 | 2500 | 2500 | 2500 | 2500 | 2500 | 1025 |
| Threads | 4 | 4 | 4 | 4 | 4 | 6 | 4 | 4 | 4 |
| Num workers | 8 | 8 | 8 | 8 | 8 | 8 | 8 | 8 | 8 |
| Actor Architecture(MLP) | [256,256] | [256,256] | [256,256] | [256,256] | [256,256] | \ | [256,256] | [256,256] | [256,256] |
| Critic Architecture(MLP) | \ | [256,256] | [256,256] | [256,256] | [256,256] | \ | [256,256] | [256,256] | [256,256] |
| Actor Learning rate | .001 | .001 | .001 | .001 | .001 | .001 | .001 | .001 | .0006 |
| Critic Learning rate | \ | .001 | .001 | .001 | .001 | \ | .001 | .001 | .0006 |
| Episode Length | 1000 | 1000 | 1000 | 1000 | 1000 | 1000 | 1000 | 1000 | 1000 |
| $\gamma$ | 1.00 | 0.99 | 0.99 | 0.99 | 0.99 | 0.99 | 0.99 | 0.99 | 0.99 |
| $\tau$ | .005 | .005 | .005 | .005 | .005 | .005 | .005 | .005 | .005 |
| $h^+$ | \ | \ | \ | \ | \ | \ | \ | \ | .2 |
| $h^-$ | \ | \ | \ | \ | \ | \ | \ | \ | .2 |
| PID | \ | [.1,.003,.001] | [.1,.003,.001] | \ | \ | \ | \ | \ | [.1,.003,.001] |
| $E$ | \ | \ | \ | \ | \ | \ | \ | \ | 4 |
| $k$ | \ | \ | \ | \ | \ | \ | \ | \ | 2.0 |
| $\alpha$ | \ | \ | \ | \ | \ | \ | \ | \ | .2 |

## C.2 Choice of Policy Class

- Standard Gaussian Policy Class: $\Pi = \{a \sim \mathcal{N}(m_\theta(s), \Sigma_\theta(s))\}$, usually the covariance $\Sigma_\theta(s)$ matrix is a diagonal matrix.

- Gaussian Policy Class with constant variance: $\Pi = \{a \sim \mathcal{N}(m_\theta(s), \sigma^2 I)\}$, here the covariance matrix $\sigma^2 I$ is state-independent.

- Dirac Policy family: $\Pi = \{a = m_\theta(s)\}$, we can approximate this as a Gaussian policy with variance close to 0.

## C.3 Training Details

As illustrated in Figure 6, our experiments are conducted in Safety–Gym environments and Bullet-Gymnasium environments. In this section, we present the training curves for each task. For every task, we run five independent seeds and plot, at each training step, the mean and standard deviation of both reward and cost.

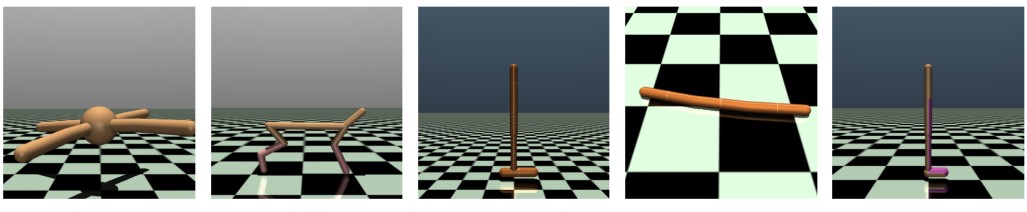

(a) SafetyAntVel  (b) SafetyHalfChee- (c) SafetyHopperVel (d)    SafetySwim- (e) SafetyWalkerVel
                      tahVel                                   merVel

Figure 6: Safety-Gym Environments

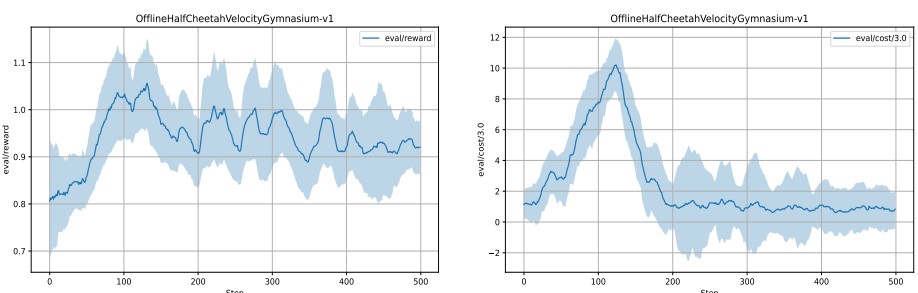

Figure 7: HalfCheetah Reward          Figure 8: HalfCheetah Cost

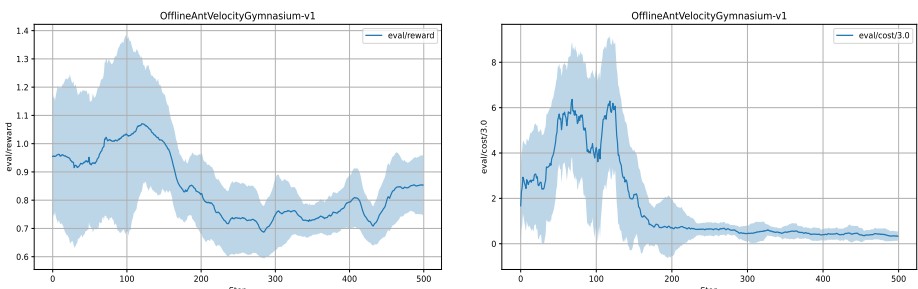

Figure 9: Ant Reward                  Figure 10: Ant Cost

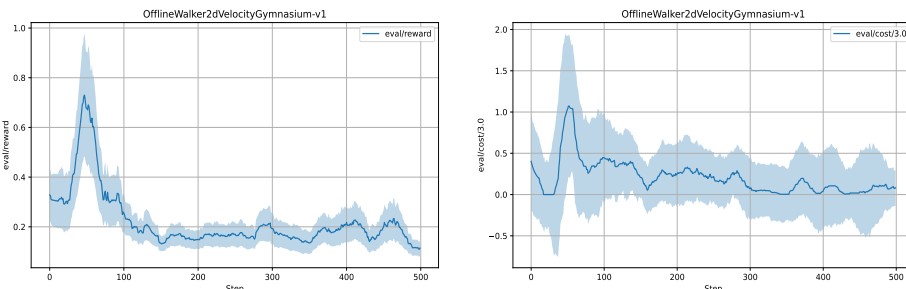

Figure 11: Walker2D Reward            Figure 12: Walker2D Cost

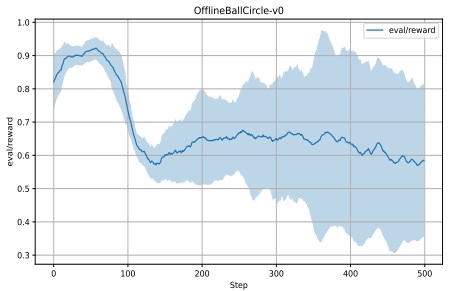

Figure 13: OfflineBallCircle Reward

Figure 14: OfflineBallCircle Cost

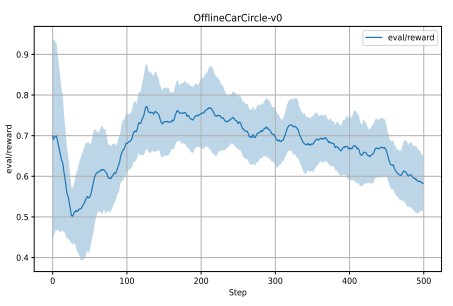
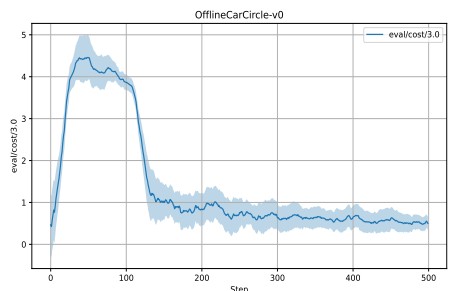

Figure 15: OfflineCarCircle Reward

Figure 16: OfflineCarCircle Cost

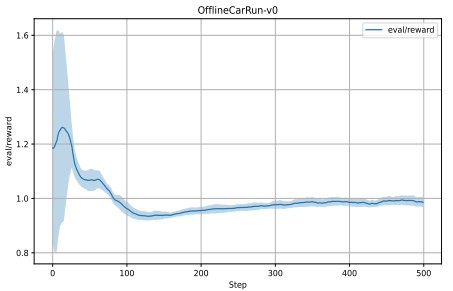
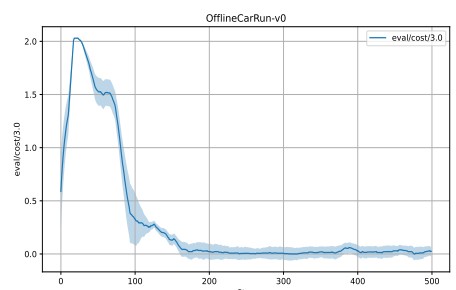

Figure 17: OfflineCarRun Reward

Figure 18: OfflineCarRun Cost

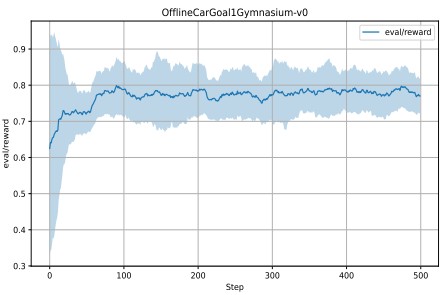
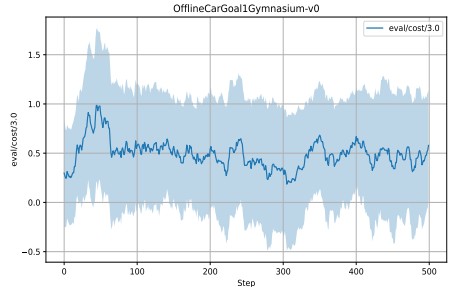

Figure 19: OfflineCarGoal1 Reward

Figure 20: OfflineCarGoal1 Cost

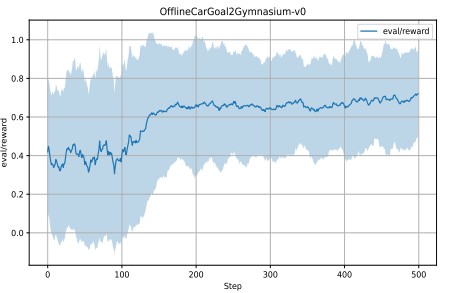

Figure 21: OfflineCarGoal2 Reward

Figure 22: OfflineCarGoal2 Cost

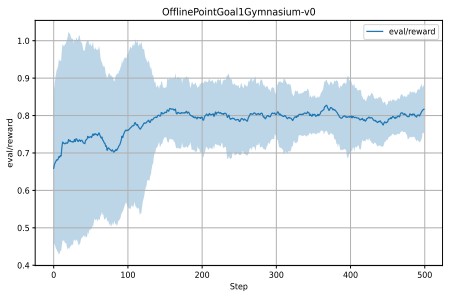
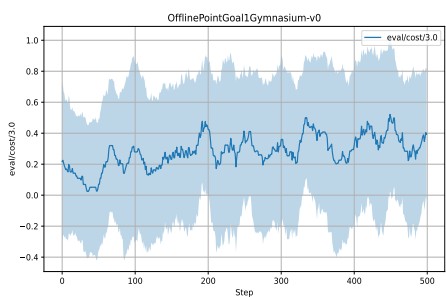

Figure 23: OfflinePointGoal1 Reward

Figure 24: OfflinePointGoal1 Cost

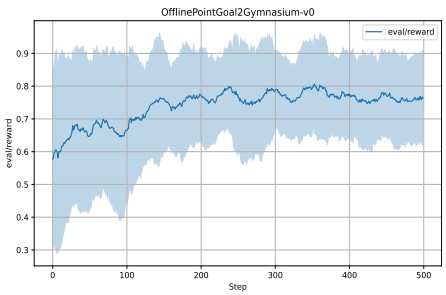
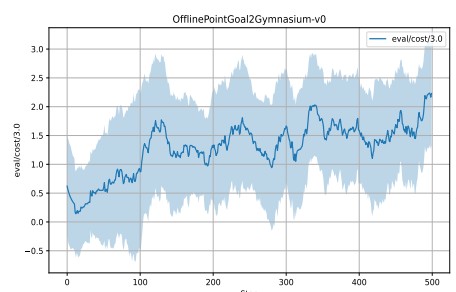

Figure 25: OfflinePointGoal2 Reward

Figure 26: OfflinePointGoal2 Cost

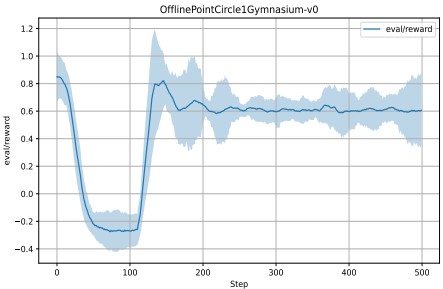
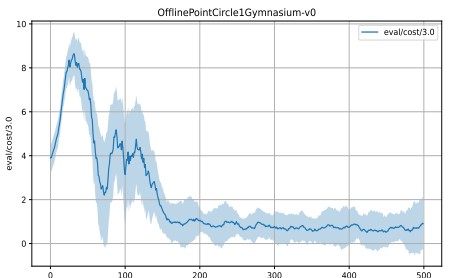

Figure 27: OfflinePointCircle1 Reward

Figure 28: OfflinePointCircle1 Cost

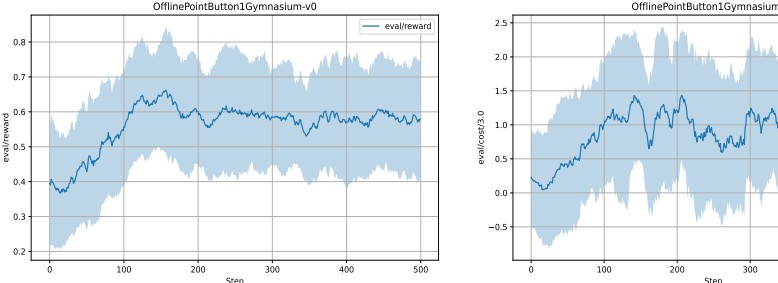

Figure 29: OfflinePointButton1 Reward  Figure 30: OfflinePointButton1 Cost

