# OpenReview forum: "Don’t Trade Off Safety: Diffusion Regularization for Constrained Offline RL"
_NeurIPS.cc/2025/Conference — NeurIPS 2025 poster_

### Official Review · Reviewer_uHfr · 2025-07-01

**Clarity:** 2
**Significance:** 3
**Originality:** 3
**Rating:** 5
**Confidence:** 3

**Summary:**

The authors propose a two-step approach for constrained offline RL, where on the first step a diffusion behavioral cloning policy is learned, which is distilled into a safe offline policy on the second step of learning. This allows to use the expressivity of diffusion to capture the behavioral policy and a faster inference of standard policies. On the second step of training the authors use a gradient manipulation approach to handle simultaneous optimization over rewards and costs. The algorithm is then validate on a standard benchmark showcasing superior performance to other baselines

**Questions:**

1. Is pre-training necessary? Is it possible to have a one stage training with diffusion regularization?
2. Proposition 3.1 and Theorem 3.1 seem to be very conservative. Personally, I wouldn’t prioritize adding these results to the main paper, but up to the authors.
2. What is BC-safe in experiments? Is it the diffusion pretrained policy? If not I recommend including it the results table
3. Why not include TREBI Lin et al. [2023] and FISOR Zheng et al. [2024] in the results table?

**Ethical Concerns:**

["NO or VERY MINOR ethics concerns only"]

**Final Justification:**

The authors alleviated some of my concerns and I trust the authors will incorporate the changes we discussed into the camera-ready version.

**Limitations:**

Limitations are properly discussed.

**Paper Formatting Concerns:**

No concerns

**Quality:**

3

**Strengths And Weaknesses:**

### Strengths
1. An interesting combination of ideas for a fast inference offline policy, for example, the modify diffusion regularisation was proposed by Chen et al. [2023], safe adaptation was proposed by Gu et al. [2024a] and the authors adapt it to the online setting, use IQL for pre-training
2. Good experimental section showing the superior performance of their method.
3. Taking care of inference costs of diffusions and proposing a method to overcome it.
### Weaknesses
1. Ablation study is rather short, I would expand it and sacrifice theoretical results
2. The authors make design choices for using diffusion vs a different BC regularisation, for diffusion optimization itself, safety adaptation vs standard lagrangian, using UCB vs conservative Q learning. Ideally, this should be properly discussed and ablated. Why did the authors make these choices is not entirely clear. But it is possible that i missed something.
3. The flow of the paper can be improved. I appreciate the idea of first going through theoretical considerations and then presenting practical algorithms, but I feel in this case it forces the reader to go back and forth between different sections. It’s not a major weakness, but a better flow would definitely sell the method better

My assessment is not final and I hope that the authors' rebuttal can alleviate some of my concerns.

---

> ### Author Rebuttal · Authors · 2025-07-28
>
> Thank you very much for your time and thoughtful comments. We sincerely appreciate your constructive feedback and provide the following clarifications to address your concerns.
> > **Is pre-training necessary? Is it possible to have a one stage training with diffusion regularization?**
>
> **Response:** We include a pre-training stage primarily to initialize the policy close to the behavioral policy, which serves as a good starting point for the subsequent regularized optimization. Notably, this pre-training phase is computationally inexpensive—accounting for only 50 out of 500 total training epochs—and is intended to stabilize early learning rather than determine final performance. While it is indeed possible to adopt a one-stage training procedure with diffusion regularization from scratch, we observe in our experiments that doing so yields only marginal differences in final performance. Thus, we choose pre-training for practical stability and simplicity without introducing unnecessary complexity to the optimization.
>
> > **Proposition 3.1 and Theorem 3.1 seem to be very conservative. Personally, I wouldn’t prioritize adding these results to the main paper, but up to the authors.**
>
> **Response:** We appreciate the reviewer’s perspective. While the theoretical results may appear conservative in form, we believe they serve two important purposes. Proposition 3.1 demonstrates that if the behavioral policy is of sufficiently high quality, then keeping the learned policy close to it (via diffusion regularization) guarantees an upper bound on the incurred cost. This motivates the design of our regularization framework and provides theoretical justification for diffusion modeling in safety-critical settings. Theorem 3.1, on the other hand, formalizes our learning guarantee: it shows that after $T$ rounds of training, the algorithm can achieve a near-optimal reward while maintaining approximate safety. Although these bounds are not tight, we believe they offer valuable insight into the structure and intuition of our approach. We will consider relocating them to the appendix if space constraints arise.
>
> > **What is BC-safe in experiments? Is it the diffusion pretrained policy? If not I recommend including it the results table.**
>
> **Response:** Thank you for the question. BC-Safe refers to a policy obtained via behavior cloning using only the safe trajectories from the offline dataset. In contrast, our diffusion-pretrained policy is trained on the entire dataset, which includes both safe and unsafe trajectories—resulting in higher cost despite achieving strong reward performance. BC-Safe is included in the results table to provide a lower-bound reference for safety-oriented imitation learning. To further clarify, we now also include the results of the pretrained policy below for comparison:
>
> | Reward (Cost) | Pretrained |
> |---------------|------------|
> | AntVel        | 0.99 (4.36) |
> | HalfCheetahVel| 0.98 (19.04) |
> | HopperVel     | 0.76 (12.14) |
> | Walker2DVel   | 0.78 (5.38) |
> | CarCircle     | 0.65 (11.16) |
> | BallCircle    | 0.72 (9.34) |
> | CarRun        | 0.96 (1.88) |
>
> This table highlights that pretraining alone does not yield a fully safe policy, thereby justifying the need for our full DRCORL training procedure with safety-aware fine-tuning.
>
> > **Why not include TREBI Lin et al. [2023] and FISOR Zheng et al. [2024] in the results table?**
>
> **Response:** We thank the reviewer for this suggestion. Regarding FISOR [Zheng et al., 2024], it is designed for a hard-constraint setting where constraint violations are not allowed at any timestep, whereas our setup focuses on soft constraints where only cumulative cost must remain below a threshold. Due to this fundamental difference in problem formulation, a direct empirical comparison is less meaningful.
>
> As for TREBI [Lin et al., 2023], we agree it is a relevant baseline as it uses diffusion models for safe planning. However, since we already include CDT—which uses transformers and trajectory-level modeling for safety adaptation—we initially opted not to include TREBI to avoid redundancy among generative model baselines. Nonetheless, we recognize its importance and will include TREBI in future versions of our experimental results to provide a more complete comparison. We appreciate the reviewer’s recommendation and will revise the experiment section accordingly.
>
> Above all, we hope that we can address your questions and we would be happy to further discuss or clarify any remaining concerns in future exchanges.

---

> > ### Author Response · Authors · 2025-08-03
> >
> > Dear Reviewer uHfr,
> >
> > Thank you again for your time and valuable feedback. As the author-reviewer discussion deadline approaches, we hope that our previous responses have addressed the concerns you raised. Please kindly let us know if there are any remaining issues or points, we would be happy to provide further details or clarifications.
> >
> > Best regards,
> >
> > The authors

---

> > > ### Comment · Reviewer_uHfr · 2025-08-03
> > >
> > > Thank you for the reminder!
> > >
> > > Re pre-training. Maybe I am missing something, but it seems that a pretraining stage is more complicated than having a one stage approach.  Can you elaborate why you think it's more difficult to have one stage instead of two?
> > >
> > > I think overlooking TREBI comparison remains a weakness of the paper.
> > >
> > > I also noticed that the authors did not comment on the highlighted weaknesses. I think it's worth adding a few sentences on the weaknesses as well.
> > >
> > > Overall, I want to know why guided different choices and evidence supporting these choices. I think the readers would be interested in that as well.

---

> > > > ### Author Response · Authors · 2025-08-04
> > > > **Reply to Reviewer uHfr's Comment**
> > > >
> > > > Dear Reviewer,
> > > >
> > > > We sincerely appreciate your active engagement and valuable feedback. Please see our detailed responses to the concerns you raised below.
> > > >
> > > >
> > > > > **Comment:** Ablation study is rather short, I would expand it and sacrifice theoretical results. The authors make design choices for using diffusion vs a different BC regularization, for diffusion optimization itself, safety adaptation vs standard Lagrangian, using UCB vs conservative Q learning. Ideally, this should be properly discussed and ablated. Why did the authors make these choices is not entirely clear. But it is possible that I missed something.
> > > >
> > > > **Response:** We appreciate this insightful point. Our theoretical results are primarily intended to provide intuition for how and why our algorithm works, rather than to dominate the empirical narrative. In future versions, we plan to expand the ablation study to cover more of our design choices, for example, comparing diffusion regularization against other behavior cloning regularizations such as $\ell^2$ loss.
> > > >
> > > > We choose diffusion regularization because it effectively penalizes the distance between the learned policy and the behavioral policy in a more expressive and distribution-aware manner (reverse-KL directly penalize the distributional distance). Specifically, diffusion models allow us to approximate the behavioral policy and efficiently compute the gradient of the reverse KL divergence.
> > > >
> > > > Regarding the safety adaptation step, we employ a gradient manipulation approach rather than a standard Lagrangian method to address the latter’s known stability issues, which often result in oscillations or unstable constraint satisfaction during training.
> > > >
> > > > Nevertheless, we fully agree that these choices should be more thoroughly discussed and empirically validated in the main text. We sincerely appreciate your insightful suggestions and will incorporate a more comprehensive ablation analysis in future revisions.
> > > >
> > > >
> > > >
> > > > > **Comment:** Re pre-training. Maybe I am missing something, but it seems that a pretraining stage is more complicated than having a one stage approach. Can you elaborate why you think it's more difficult to have one stage instead of two?
> > > >
> > > > **Response:** Thank you for raising this valuable point. We agree that a one-stage training approach might appear simpler conceptually. However, in our case, the main complication arises from jointly updating the diffusion policy and the learned policy within the same training loop.
> > > >
> > > > In particular, the goal of diffusion regularization is to constrain the learned policy to remain close to the behavioral policy. If the diffusion model is being updated simultaneously with the policy, we lose the ability to rely on a stable reference distribution during early training. This can lead to instability, especially at the beginning, when both models are still learning and evolving.
> > > >
> > > > By contrast, our pretraining stage ensures that the diffusion model has already approximated the behavioral policy before we begin optimizing the learned policy. This allows us to apply diffusion-based regularization in a stable and effective manner throughout the main training process. While one-stage training may still work in principle, we found that this two-stage approach improves training stability and overall performance.
> > > >
> > > >
> > > >
> > > >
> > > > > **Comment:** I think overlooking TREBI comparison remains a weakness of the paper.
> > > >
> > > > **Response:** Thank you for pointing this out. We fully agree that TREBI is a representative work in the area of offline Safe RL, as it also leverages diffusion models for planning and performs safety adaptation using trajectory-level information. We have cited TREBI in the Related Work section under the literature on diffusion models in offline RL.
> > > >
> > > > The reason we did not include TREBI in our experimental baselines is that we had already included Constrained Decision Transformer (CDT), which also utilizes trajectory-level information for safety-aware decision-making. At the time, we viewed CDT as a comparable counterpart in terms of generative modeling and trajectory inference.
> > > >
> > > > That said, we acknowledge the value of directly comparing with TREBI and will aim to include it in our experimental baselines in future versions of the paper. We appreciate your suggestion and agree that such a comparison would further strengthen the empirical evaluation.
> > > >
> > > >
> > > > Overall, we sincerely appreciate your participation in the discussion and your insightful comments. Your feedback has been very helpful, and we hope to use it to further improve the paper and contribute meaningfully to the field of Safe RL.

---

> > > > > ### Comment · Reviewer_uHfr · 2025-08-05
> > > > >
> > > > > Thank you for clarifications. I think parts of our discussion should make it's way to the paper if it's accepted.
> > > > >
> > > > > While I find the weaknesses raised by Reviewer 5D62 important, in my opinion the paper is still closer to acceptance than rejection and will raise the score based on our discussion.

---

> ### Author Response · Authors · 2025-08-05
>
> Dear Reviewer uHfr,
>
>
> Thank you very much for your thoughtful feedback and for engaging in the discussion. We truly appreciate your constructive comments, which have helped us better understand how to improve the clarity and completeness of our work. Your insights have been instrumental in refining both our presentation and the broader framing of the contributions.
>
> In particular, we acknowledge the importance of expanding our ablation study to more thoroughly evaluate the impact of individual design choices, as well as improving the coverage of baseline comparisons, especially for literature like TREBI leveraging diffusion model for planning. These are excellent suggestions, and we are committed to addressing them in future revisions of the paper.
>
> We’re very grateful for your support and thoughtful participation throughout the review process.

---

### Official Review · Reviewer_Fxxd · 2025-07-02

**Clarity:** 3
**Significance:** 3
**Originality:** 3
**Rating:** 5
**Confidence:** 3

**Summary:**

This paper proposes a diffusion-regularized method for constrained offline RL. The key idea is to use a diffusion model to approximate the behavioral policy from offline data, then distill it into a lightweight Gaussian policy that enables fast inference. A gradient manipulation mechanism is further introduced to trade off between maximizing reward and minimizing constraint violations. The authors evaluate the method across DSRL benchmarks and claim superior performance in both reward and safety metrics compared to existing safe offline RL baselines.

**Questions:**

* Why use diffusion policies at all? Wouldn’t other generative models work just as well or better?
* Is there any study of what happens when the diffusion model fails to fully capture the behavioral policy? How does this affect safety guarantees?
* The method applies reverse KL regularization. Did the authors try forward KL or a symmetric variant?
* The theoretical analysis drops the KL regularization—how valid are the bounds in practical settings where the reverse KL is strong?

**Ethical Concerns:**

["NO or VERY MINOR ethics concerns only"]

**Quality:**

3

**Strengths And Weaknesses:**

## Quality
The pipeline is solid, and the empirical results are comprehensive. The gradient manipulation approach is a reasonable adaptation of ideas. The paper also provided convergence analysis. However, I didn't check the correctness of the theories.

However, I’m still unclear what the diffusion model is doing beyond replacing a behavior cloning model. Given that the final policy is a simple Gaussian actor and inference doesn’t sample from the diffusion trajectory, what exactly does the diffusion model help with—other than regularizing the Gaussian policy via score matching? The paper assumes the diffusion model perfectly recovers the behavior policy, but in practice this is far from guaranteed, especially in high-dimensional action spaces. That assumption weakens the theoretical safety bound.

Also, learning from the diffusion model simulated behavioral policy raises concerns about compounding error. If the score function is misestimated, wouldn't the final policy be regularized toward the wrong directions?

## Clarity
The paper is mostly well written, though the presentation gets dense in the methodology section. But the paper could benefit from a cleaner separation between components: what is learned in pretraining, what is used during policy optimization, and what is actually deployed at test time.

## Novelty
The use of diffusion models as offline regularizers is not new. Prior work already explore this line. The key novelty here is combining the regularization with a soft gradient-switching mechanism adapted to safety-constrained offline RL.

Also, why using diffusion policy at all? If the goal is to regularize the final policy toward the offline data distribution, many simpler generative models (e.g., VAE, normalizing flow) could be used with faster training and easier analysis. The authors argue diffusion models have stronger generative capacity—but that strength is underutilized here since they only extract scores, not full samples. Some justification is needed.

## Significance
The method works. Empirically it delivers strong reward under safety constraints, and the lightweight policy at inference is indeed faster than diffusion-based baselines like TREBI or FISOR. So from an engineering point of view, this is a practical and deployable method.

---

> ### Author Rebuttal · Authors · 2025-07-28
>
> Thank you very much for your time and thoughtful comments, we totally recognize the reviewer's contributions for the comments, and we would like to give credits to the reviewers. We will reflect on the reviewer's comments and improve our work based on the suggestions in the future. Here we would also like to provide the following clarifications to address the concerns you raise.
>
> > **Question:** Why use diffusion policies at all? Wouldn’t other generative models work just as well or better?
>
> **Response:** We choose diffusion policies for two key reasons. First, diffusion models are highly expressive and data-efficient, as they can approximate complex, multi-modal behavior distributions using relatively limited training data. This is particularly important in offline RL settings, where data is fixed and may be limited or unevenly distributed.
>
> Second, diffusion models offer a computational advantage via score-based learning. Specifically, we leverage the score function to estimate the gradient of the reverse KL regularization term without needing to sample trajectories from the generative process. This avoids the high computational cost typically associated with sampling in diffusion models.
>
> While other generative tools like VAEs are also theoretically applicable, they require sampling multiple actions and computing expectations over the log-likelihood gradients, which can be less efficient and more prone to high variance. Moreover, in our experiments, diffusion-based regularization consistently yielded better empirical performance, particularly in terms of safety and stability compared to variational approaches we have tried in preliminary ablation studies.
>
> Nevertheless, we do acknowledge that other generative approaches, including VAEs or flow-based models, may offer alternative trade-offs in modeling and optimization. For example, [2] applies constrained VAEs in safe RL, which we consider a promising future direction for extension and comparison.
>
>
> > **Question:** Is there any study of what happens when the diffusion model fails to fully capture the behavioral policy? How does this affect safety guarantees?
>
> **Response:** We acknowledge that when the offline dataset is sparse or highly heterogeneous, diffusion models may fail to accurately approximate the behavioral policy. While existing works such as TREBI [3] and OASIS [4] implicitly assume that diffusion models are sufficiently expressive to model the dataset distribution, to our knowledge, few studies have systematically analyzed what happens when this assumption breaks.
>
> In our framework, if the diffusion policy fails to match the behavioral policy, the main impact is on the initial policy distribution and the regularization signal during training. However, DRCORL is designed to be robust to this mismatch: our safety adaptation procedure (outlined in the main algorithm) fine-tunes the learned policy while respecting safety constraints. Thus, even under imperfect modeling, the algorithm can still converge to a safe policy, though we may also encounter potential degraded performance.
>
> > **Question:** The method applies reverse KL regularization. Did the authors try forward KL or a symmetric variant?
>
> **Response:** We appreciate the reviewer’s question. We chose reverse KL regularization because of its mode-seeking nature, which is well-suited for constraining the learned policy near the behavioral policy,  especially when the dataset contains multiple modes. Reverse KL tends to avoid assigning probability mass to unsupported regions, thereby reducing the risk of unsafe generalization.
>
> This design choice is also motivated by prior work, including [1], which analyzes the use of forward vs. reverse KL in policy optimization. We agree that symmetric divergences or other variants (e.g., Jensen-Shannon or $\alpha$-divergence) are worth exploring and plan to investigate them in future work.
>
> > **Question:** The theoretical analysis drops the KL regularization—how valid are the bounds in practical settings where the reverse KL is strong?
>
> **Response:** This is an insightful question. We omit the KL regularization in the theoretical analysis to simplify the derivation and make the regret bounds more interpretable. However, in practice, the inclusion of the reverse KL term plays a critical role in stabilizing training and encouraging conservative policy updates.
>
> Our assumption is that the optimal policy lies in the neighborhood of the behavioral policy. Under this assumption, the solutions to the optimization problems, with or without KL regularization, will be close. Hence, while the bounds do not formally incorporate the KL term, we believe they still meaningfully capture the learning behavior of the algorithm in realistic settings where the reverse KL penalty is used.
>
> Above all, we hope that we have addressed your questions. We greatly value your insights and would be happy to further discuss or clarify any remaining concerns in future exchanges.
>
> **References:**
>
> [1] Chan, Alan, et al. "Greedification operators for policy optimization: Investigating forward and reverse kl divergences." Journal of Machine Learning Research 23.253 (2022): 1-79.
> [2] Liu, Zuxin, et al. "Constrained variational policy optimization for safe reinforcement learning." *International Conference on Machine Learning*. PMLR, 2022.
> [3] Lin, Qian, et al. "Safe offline reinforcement learning with real-time budget constraints." *International Conference on Machine Learning*. PMLR, 2023.
> [4] Yao, Yihang, et al. "Oasis: Conditional distribution shaping for offline safe reinforcement learning." *Advances in Neural Information Processing Systems* 37 (2024): 78451-78478.

---

> > ### Author Response · Authors · 2025-08-03
> >
> > Dear Reviewer Fxxd,
> >
> > Thank you again for your time and valuable feedback. As the author-reviewer discussion deadline approaches, we hope that our previous responses have addressed the concerns you raised. Please kindly let us know if there are any remaining issues or points, we would be happy to provide further details or clarifications.
> >
> > Best regards,
> >
> > The authors

---

> ### Comment · Area_Chair_2zXg · 2025-08-05
> **Please engage with the authors' response.**
>
> Dear reviewer QXQe,
>
> Thanks for your reviewing efforts so far. Please engage with the authors' response.
>
> Thanks,
> Your AC

---

### Official Review · Reviewer_5D62 · 2025-07-02

**Clarity:** 2
**Significance:** 2
**Originality:** 2
**Rating:** 4
**Confidence:** 4

**Summary:**

The paper proposed Diffusion-Regularized Constrained Offline Reinforcement Learning (DRCORL), which uses a diffusion model to capture the behavioral policy from offline data and then extracts a simplified policy to enable efficient inference. The diffusion model is used to compute the score function of offline data and used to compute policy gradient with reverse KL divergence regularization. Then the gadient maniputation tricks in Gu et, al. is used to update the constraint-satisfying policies.

**Questions:**

1. Why not use other score matching techniques mentioned in weaknesses 2 to learn the score function? I don't see any advantages in using the denoising score matching as the noise perturbation design is mainly beneficial to the generation process of diffusion models. Especially, the denoising score matching learns the score functions of the noise-perturbed data rather than the true score.

2. How to compute the $t\to 0$ in line 237?

3. The pretrained model can only guaratee learning the score function of the expert policy accurately on the dataset distribution, but not on the current behavior policy distribution. How to handle these distributional shift when you query the score function of current policy using score network trained from the offline dataset?

4. I might not be fully aware of the thoeritical contributions as no siginifcant parameters show up in the theoritical results. How is the theoritical results compared to other baseline algorithms, or none of the baseline algorithms can have such theoritical guarantees?

**Ethical Concerns:**

["NO or VERY MINOR ethics concerns only"]

**Final Justification:**

I have read the comments from other reviewers and AC. I will raise my initial score since the author's rebuttal resovled by major concern.

**Limitations:**

Yes

**Quality:**

2

**Strengths And Weaknesses:**

## Strength
1. The writing is clear.
2. comprehensive evaluations

## Weakness
1. The reverse KL regularization is not novel and has been studied for RL algorithms such as [1].
2. The authors only learns the score function, which has connections to but is not fully equivalent to the claimed diffusion models, that is mainly for generation. There are a lot of alternative approaches to learn such score functions, such as [2, 3], so the diffusion model is not a significant component here. The title might be misleading.
3. The theoretical results are a bit general.
4. The main algorithm directly follows an online RL algorithm in Gu et, al. Adding regularization in offline RL is a standard technique when converting online RL to offline.

[1] Chan, Alan, et al. "Greedification operators for policy optimization: Investigating forward and reverse kl divergences." Journal of Machine Learning Research 23.253 (2022): 1-79.
[2] Hyvärinen, Aapo, and Peter Dayan. "Estimation of non-normalized statistical models by score matching." Journal of Machine Learning Research 6.4 (2005).
[3] Song, Yang, et al. "Sliced score matching: A scalable approach to density and score estimation." Uncertainty in artificial intelligence. PMLR, 2020.

---

> ### Author Rebuttal · Authors · 2025-07-28
>
> Thank you very much for your time and thoughtful comments. We sincerely appreciate your constructive feedback and provide the following clarifications to address your concerns.
>
>
> > **Question:** Why not use other score matching techniques mentioned in weaknesses 2 to learn the score function? I don't see any advantages in using the denoising score matching as the noise perturbation design is mainly beneficial to the generation process of diffusion models. Especially, the denoising score matching learns the score functions of the noise-perturbed data rather than the true score.
>
> **Response:** We thank the reviewer for raising this important point, we think the reviewer is asking about why do we need to use the noise perturbation to learn the score function, as mentioned in weaknesses2 we do not have to utilize diffusion model to learn the score function.
>
> It is important to emphasize that the core role of the diffusion model in our framework is to approximate the behavioral policy, and we use this approximation to regularize the learned policy. The score function derived from the diffusion model is a means to this end, as it serves as a tractable surrogate for imposing a constraint that keeps the learned policy not far away from the behavioral one. We would like to clarify that our use of the score function primarily serves a practical purpose: it allows us to compute the gradient of the reverse KL divergence in Eq.(8) without needing to sample from the diffusion model, which is often computationally expensive. This significantly improves efficiency during policy optimization.
>
> We can also compute the gradient of the reverse-KL by utilizing the diffusion model to sample multiple actions, then take the gradient of the estimator as the unbiased estimated gradient. The use of the score function simply serves to  improve the computation efficiency in the training process without having to use the diffusion model for sampling.
>
>
>
> In summary, while we utilize the score function for computational reasons, the underlying regularization mechanism still relies on the diffusion model’s ability to capture the structure of the behavioral policy. We appreciate the reviewer’s suggestion regarding alternative score matching methods and will consider exploring more accurate or efficient options in future work.
>
>
> > **Question:** How to compute the $t\to 0$ in line 237?
>
> **Response:** In our practical implementation, we discretize the$\frac{1}{\sqrt{\bar{\beta_t}}} \epsilon_{\psi}(a_t,t\vert s)$ in Line 237 as an approximation.
>
> ---
>
> > **Question:** The pretrained model can only guarantee learning the score function of the expert policy accurately on the dataset distribution, but not on the current behavior policy distribution. How to handle these distributional shift when you query the score function of current policy using score network trained from the offline dataset?
>
> **Response:** We acknowledge that the score network is trained on the empirical distribution of the offline dataset, which may differ from the current policy’s distribution. However, as stated in Theorem 3.1, we assume the dataset is sufficiently large such that the empirical distribution is close to the behavioral policy. This is reflected in the small $\epsilon_{\text{offline}}$, which appears explicitly in our regret bounds in Eq.(14) and Eq.(15), quantifying the distributional mismatch. Therefore, our theoretical analysis directly accounts for this issue and bounds its impact.
>
> > **Question:** I might not be fully aware of the theoretical contributions as no significant parameters show up in the theoretical results. How is the theoretical results compared to other baseline algorithms, or none of the baseline algorithms can have such theoretical guarantees?
>
> **Response:** We appreciate the reviewer’s perspective on the theoretical contributions. While the results may not involve complex parameter dependencies, we highlight several key contributions:
>
> 1. Regret Bounds for Both Reward and Cost: Theorem 3.1 provides bounds on both the expected reward and constraint violation (cost) of the learned policy compared to the optimal one, which is relatively rare in the safe offline RL literature.
> 2. Cost Upper Bound via Diffusion Regularization: Proposition 3.1 demonstrates that pretraining a diffusion model and regularizing the learned policy to stay near the behavioral policy helps upper bound the cost. This highlights the importance of diffusion-based regularization in safe RL.
>
> Compared to existing works:
> - Methods such as TREBI [1], OASIS [2], and CVPO [3] do provide regret bounds, but they are often based on strong assumptions, and some conclusions are not particularly significant or tight in practice.
> - FISOR [4] and CDT [5] only offer a closed-form representation of the optimal solution without any accompanying theoretical guarantees for the performance of the learned policy.
> - CRPO [6] and CPO [7] do include theoretical guarantees, but they are established in an online Safe RL setting, making them inapplicable to the offline setting we study.
>
> In summary, our theoretical contributions are tailored to the offline safe RL context, and we believe they fill a meaningful gap by providing both regret and constraint violation guarantees without relying on strong assumptions from the online literature. We thank the reviewer for highlighting this point and will aim to further expand the theoretical scope in future versions.
>
>
> **References:**
>
> [1] Lin, Qian, et al. "Safe offline reinforcement learning with real-time budget constraints." *International Conference on Machine Learning*. PMLR, 2023.
>
> [2] Yao, Yihang, et al. "Oasis: Conditional distribution shaping for offline safe reinforcement learning." *Advances in Neural Information Processing Systems* 37 (2024): 78451-78478.
>
> [3] Liu, Zuxin, et al. "Constrained variational policy optimization for safe reinforcement learning." *International Conference on Machine Learning*. PMLR, 2022.
>
> [4] Zheng, Yinan, et al. "Safe offline reinforcement learning with feasibility-guided diffusion model." *arXiv preprint arXiv:2401.10700* (2024).
>
> [5] Liu, Zuxin, et al. "Constrained decision transformer for offline safe reinforcement learning." *International Conference on Machine Learning*. PMLR, 2023.
>
> [6] Xu, Tengyu, Yingbin Liang, and Guanghui Lan. "Crpo: A new approach for safe reinforcement learning with convergence guarantee." *International Conference on Machine Learning*. PMLR, 2021.
>
> [7] Achiam, Joshua, et al. "Constrained policy optimization." *International Conference on Machine Learning*. PMLR, 2017.

---

> > ### Author Response · Authors · 2025-08-03
> >
> > Dear Reviewer 5D62,
> >
> > Thank you again for your time and valuable feedback. As the author-reviewer discussion deadline approaches, we hope that our previous responses have addressed the concerns you raised. Please kindly let us know if there are any remaining issues or points, we would be happy to provide further details or clarifications.
> >
> > Best regards,
> >
> > The authors

---

> > > ### Comment · Reviewer_5D62 · 2025-08-04
> > >
> > > Thanks for your detailed reply and explanation. Now I understand the paper's contribution more clearly.
> > >
> > > I agree that learning the score function also means using the diffusion model to capture the data distribution, which plays an important role in computing the gradient of reverse KL regularization.
> > >
> > > Nonetheless, I suggest that the author should carefully revise the writing to explain the use of the score function and why it is a better way to leverage the diffusion model in offline RL.

---

> ### Author Response · Authors · 2025-08-04
>
> Dear Reviewer 5D62,
>
>
> Thank you for this insightful comment. We recognize that certain parts of the original manuscript may have caused confusion regarding the role of the score function, potentially making it harder for readers to fully understand its purpose within our framework. As you suggested, we agree that the use of the score function should be more clearly explained as a means of reducing the computational burden during training.
>
> Based on your comment, we plan to add a clarification at the beginning of Section 3.2 (Diffusion Regularization) to explicitly explain the use of the score function and better highlight the intuition behind diffusion-based policy regularization. Additionally, we intend to include comparisons between diffusion regularization and other commonly used techniques in offline RL, such as the $\ell_2$ penalty, to more clearly illustrate why diffusion modeling is particularly suitable in this context.
>
> We sincerely appreciate your engagement in the discussion, and we believe your suggestion will significantly enhance the clarity and impact of our presentation.

---

> > ### Comment · Reviewer_5D62 · 2025-08-05
> >
> > Thanks for the reply. I will adjust my score as my major concern has been addressed.

---

> > > ### Author Response · Authors · 2025-08-05
> > >
> > > Dear Reviewer 5D62,
> > >
> > > Thank you for recognizing our efforts and for raising your score. Your thoughtful feedback has been highly valuable in helping us improve our work.  Wishing you a great day!
> > >
> > > Best regards,
> > >
> > > The authors

---

### Official Review · Reviewer_QXQe · 2025-07-04

**Clarity:** 3
**Significance:** 3
**Originality:** 3
**Rating:** 4
**Confidence:** 4

**Summary:**

This paper introduces Diffusion-Regularized Constrained Offline Reinforcement Learning (DRCORL), a framework designed to learn safe and high-performance policies from offline datasets. The approach leverages a diffusion model to imitate the behavioral policy and uses reverse KL divergence to regularize a simplified Gaussian policy. In addition, it employs a gradient manipulation technique that adaptively interpolates between competing objectives. The method is evaluated on the DSRL benchmark, outperforming several existing offline safe RL methods in more than half of the tasks.

**Questions:**

- Could you comment more on DRCORL against non-diffusion-based offline RL approaches that are safety-aware but use simpler regularization (e.g., model ensembles or cost-conditioned Q-learning)? I would like to learn more about justifying diffusion modeling as necessary?

- How does the diffusion model behave in cases where the offline dataset is sparse, multi-modal, or contains unsafe behaviors? Can the method robust enough to handle these situations which are more realistic in the real world, highlighted by some papers like Learning from Sparse Offline Datasets via Conservative Density Estimation. I have some concerns that the proposed method may be unstable when working on sparse data set.

**Ethical Concerns:**

["NO or VERY MINOR ethics concerns only"]

**Final Justification:**

Thank the authors for detailed replies. It addresses my concerns. After reviewing the replies of the authors to me and other reviewers, I would like to keep my score.

**Limitations:**

Yes

**Quality:**

3

**Strengths And Weaknesses:**

Strengths

- DRCORL employs a diffusion model to learn the complex behavioral policy from offline data, then extracts and regularizes a separate, simplified Gaussian policy using the diffusion model's score function. This design enables fast inference speeds.

- The gradient manipulation technique offers an adaptive solution to the reward-safety trade-off, making the method more generalizable across different cost constraint conditions.

- Extensive evaluations on 12 benchmark tasks from SafetyGym, BulletSafetyGym, and MuJoCo demonstrate that DRCORL achieves better performance while maintaining safety constraints, providing competitive scores against BCQ-Lag, CPQ, and COptiDICE.

Weaknesses:

- Not enough discussion or comparison of non-diffusion-based offline safe RL methods, such as ensemble-based uncertainty estimation or constrained supervised learning approaches, which may offer competitive performance without relying on generative models. Including these could better contextualize the advantages of DRCORL.

- While DRCORL avoids inference-time sampling from the diffusion model, pretraining remains computationally expensive, especially for large-scale problems. In high-dimensional state-action spaces (e.g., manipulation with vision inputs or multi-agent systems), training diffusion models and critics can become very resource-intensive, limiting scalability and practical deployment. In addition, it may require a high-quality offline dataset and relies on good coverage of the behavioral policy to generate the diffusion models. Its performance may degrade under poor dataset conditions.

---

> ### Author Rebuttal · Authors · 2025-07-28
>
> Thank you for engaging in the discussion and providing further clarification. We sincerely appreciate the reviewer’s thoughtful and constructive suggestions, which have helped improve the clarity and generalization perspective of our work. Your feedback has been constructive in strengthening our argument for DRCORL and its broader applicability.
>
> > **Question:** Could you comment more on DRCORL against non-diffusion-based offline RL approaches that are safety-aware but use simpler regularization (e.g., model ensembles or cost-conditioned Q-learning)? I would like to learn more about justifying diffusion modeling as necessary?
>
> **Response:** Thank you for raising this insightful question. In fact, several baselines included in our experiments are safety-aware and employ simpler regularization techniques. For example, CAPS [1] uses an $\ell^2$ penalty (see Eq.(6)) to constrain the learned policy to remain close to the behavior policy, while BEAR [2] employs bootstrapping and MMD-based penalties to avoid out-of-distribution (OOD) actions. These approaches, like ours, share the core objective of regularizing the learned policy within a trust region around the behavior policy, which is a widely adopted strategy in offline RL.
>
> What differentiates our approach is the use of diffusion models to represent this regularization constraint. Diffusion models possess strong approximation capabilities, capable of modeling complex and multi-modal behavior distributions in a nonparametric and data-driven manner. In contrast to manually designed distance-based penalties, the diffusion-based approach learns the behavioral distribution directly and provides a more flexible and expressive way to regularize the learned policy. This allows DRCORL to scale more naturally to settings where behavior distributions are non-trivial.
>
> > **Question:** How does the diffusion model behave in cases where the offline dataset is sparse, multi-modal, or contains unsafe behaviors? Can the method be robust enough to handle these situations which are more realistic in the real world, highlighted by some papers like Learning from Sparse Offline Datasets via Conservative Density Estimation. I have some concerns that the proposed method may be unstable when working on sparse data set.
>
> **Response:** This is an important and valid concern. At present, we have evaluated DRCORL on a broad set of tasks from the DSRL benchmark, which primarily features uni-modal behavior distributions. We acknowledge that more realistic scenarios, such as multi-modal or sparse datasets with heterogeneous and potentially unsafe behavior can pose significant challenges. Evaluating DRCORL under such conditions is an active direction for future work. In particular, we are interested in testing vision-based or partially observed settings, where the data may naturally exhibit multi-modality.
>
> Regarding robustness to unsafe behaviors, the DSRL benchmark does include both safe and unsafe trajectories. Our experimental results show that DRCORL is able to learn safe policies despite the presence of unsafe samples. For example, in the BC-Safe baseline, imitation learning is conducted solely on safe trajectories, yet the resulting policies are often less safe than those learned by DRCORL. This empirical finding supports the robustness of our approach in handling mixed-quality data.
>
> That said, we do recognize that the performance of our method is inherently tied to the quality and size of the offline dataset. Theoretical results (see [3]) assume that the diffusion model can successfully approximate the behavioral distribution given sufficient data. However, in highly sparse or low-quality datasets, such as those addressed in the conservative density estimation literature, this assumption may no longer hold, and more specialized solutions may be needed. We appreciate the reviewer’s pointer to this line of work and see it as a promising extension direction for improving the robustness of DRCORL in real-world deployments.
>
> **References:**
>
> [1] Chemingui, Yassine, et al. "Constraint-adaptive policy switching for offline safe reinforcement learning." *Proceedings of the AAAI Conference on Artificial Intelligence*. Vol. 39. No. 15. 2025.
>
> [2] Kumar, Aviral, et al. "Stabilizing off-policy Q-learning via bootstrapping error reduction." *Advances in Neural Information Processing Systems* 32 (2019).
>
> [3] De Bortoli, Valentin. "Convergence of denoising diffusion models under the manifold hypothesis." *arXiv preprint* arXiv:2208.05314 (2022).

---

> > ### Author Response · Authors · 2025-08-03
> >
> > Dear Reviewer QXQe,
> >
> > Thank you again for your time and valuable feedback. As the author-reviewer discussion deadline approaches, we hope that our previous responses have addressed the concerns you raised. Please kindly let us know if there are any remaining issues or points, we would be happy to provide further details or clarifications.
> >
> > Best regards,
> >
> > The authors

---

> ### Comment · Area_Chair_2zXg · 2025-08-05
> **Please engage with the authors' response.**
>
> Dear reviewer QXQe,
>
> Thanks for your reviewing efforts so far. Please engage with the authors' response.
>
> Thanks,
> Your AC

---

### Note · Authors · 2025-08-13

We thank all reviewers and the AC for their constructive feedback and active engagement.

Our work proposes Diffusion-Regularized Constrained Offline Reinforcement Learning (DRCORL), which models the behavioral policy from offline data using a diffusion model and applies gradient manipulation for safety adaptation. This design enables reliable safety performance, fast inference, and strong reward outcomes in offline safe RL, consistently satisfying cost limits and performing well across DSRL tasks with fixed hyperparameters. Compared to prior safe offline RL methods, DRCORL combines theoretical guarantees with practical applicability.

During the rebuttal, several key concerns have been addressed:
- Choice of diffusion. Selected for stability and empirical performance, diffusion models offer clear advantages over alternatives (e.g., VAEs). We use the score function to estimate the gradient without sampling from the diffusion model, improving computational efficiency. The main reason for using diffusion is its ability to accurately approximate the behavioral policy distribution.

- Two-stage vs. one-stage training. Pretraining first fixes the behavioral distribution, stabilizing early learning before policy optimization. While effective, merging the two phases is a feasible option in spite of initial training instability.

-  Theoretical results. We provide regret and cost bounds tailored to offline safe RL, complementing empirical results and offering stronger guarantees than much of the existing literature.

- Robustness and limitations. The algorithm tolerates unsafe trajectories, however, its performance is inevitably influenced by the quality of the original offline dataset. In cases where the dataset lacks sufficient safe coverage or contains strong bias, constraint satisfaction and reward performance may degrade.

- Signal from the discussion. two reviewers (5D62 and uHfr) have explicitly expressed their concerns have been addressed, with reviewer 5D62 explicitly expressing the intent to raise the score and reviewer uHfr maintained a positive score in favor of acceptance.

Overall, DRCORL fills a gap in offline safe RL, achieving high reward and safe policy learning. We sincerely thank the reviewers for their valuable suggestions, which have guided to improve our work.

---

### Decision · Program_Chairs · 2025-09-17

**Decision:**

Accept (poster)

**Comment:**

This paper proposes a safety-based offline RL approach that consists of a diffusion policy pretraining stage and constraint-based adaptation stage.

The reviewers held mostly positive opinions on the paper, but there were a variety of remaining concerns about the novelty, as it seemed to be a combination of existing ideas. As stated by Reviewer Fxxd, "The key novelty here is combining the regularization with a soft gradient-switching mechanism adapted to safety-constrained offline RL."

Overall I recommend acceptance as a poster, although I think a case could be made that the contributions aren't significant enough for acceptance.